# Robust Similarity Learning with Difference Alignment Regularization

**Shuo Chen[1], Gang Niu[1], Chen Gong[3], Okan Koc[1], Jian Yang[3], Masashi Sugiyama[1,2]**
[1]RIKEN Center for Advanced Intelligence Project, Tokyo, Japan
[2]The University of Tokyo, Tokyo, Japan
[3]Nanjing University of Science and Technology, Nanjing, China

## Abstract

Similarity-based representation learning has shown impressive capabilities in both supervised (e.g., *metric learning*) and unsupervised (e.g., *contrastive learning*) scenarios. Existing approaches effectively constrained the *representation difference* (i.e., the disagreement between the embeddings of two instances) to fit the corresponding (pseudo) similarity supervision. However, most of them can hardly restrict the variation of representation difference, sometimes leading to *overfitting* results where the clusters are disordered by drastically changed differences. In this paper, we thus propose a novel *difference alignment regularization* (DAR) to encourage all representation differences between inter-class instances to be as close as possible, so that the learning algorithm can produce *consistent* differences to distinguish data points from each other. To this end, we construct a new *cross-total-variation* (CTV) norm to measure the divergence among representation differences, and we convert it into an equivalent *stochastic form* for easy optimization. Then, we integrate the proposed regularizer into the empirical loss for *difference-aligned similarity learning* (DASL), shrinking the *hypothesis space* and alleviating overfitting. Theoretically, we prove that our regularizer tightens the error bound of the traditional similarity learning. Experiments on multi-domain data demonstrate the superiority of DASL over existing approaches in both supervised metric learning and unsupervised contrastive learning tasks.

## 1 Introduction

Recently, representation learning with pairwise similarity has demonstrated great effectiveness on many different types of data (Weinberger et al., 2006; Zadeh et al., 2016; Xu et al., 2022; Wu et al., 2022). In this problem setting, we usually consider the pairwise relationship (e.g., similar or dissimilar) between instances from the training data, and we aim to learn a generalizable feature representation to predict the pairwise similarity for the test data. This topic of *similarity learning* has already attracted much attention and has also been applied to many specific tasks such as classification, verification, and clustering (Jing & Tian, 2020; Kou et al., 2022; Zhong et al., 2020).

In the supervised case, the similarity labels of training data are usually provided by humans. This supervised setting is also referred to as *metric learning* (Xing et al., 2002), where a distance metric is learned to measure the pairwise similarity between instances in both the training and test phases. Although some existing approaches can successfully learn the corresponding feature representation of similarity metric via the nonlinear *convolutional neural networks* (CNNs) (a.k.a. *SiameseNet* (Kaya & Bilge, 2019)), the correctly predicted distances will not be penalized any more in their *empirical loss* (e.g., *triplet loss* (Hoffer & Ailon, 2015) and *ProxyAnchor loss* (Kim et al., 2020b)). This means that these *inter-class* distances/differences will vary in an underconstrained range and thus may lead to the unstable predictions that affect the model generalization on the unseen test data.

In the unsupervised (self-supervised) case, the positive (similar) pairs and negative (dissimilar) pairs are typically generated by the pseudo supervision, e.g. the *instance discrimination* (Dosovitskiy et al., 2014) and *data augmentation* (Chen et al., 2020) used in *contrastive learning* to build negative

---

[†]Correspondence to Shuo Chen (shuo.chen.ya@riken.jp).

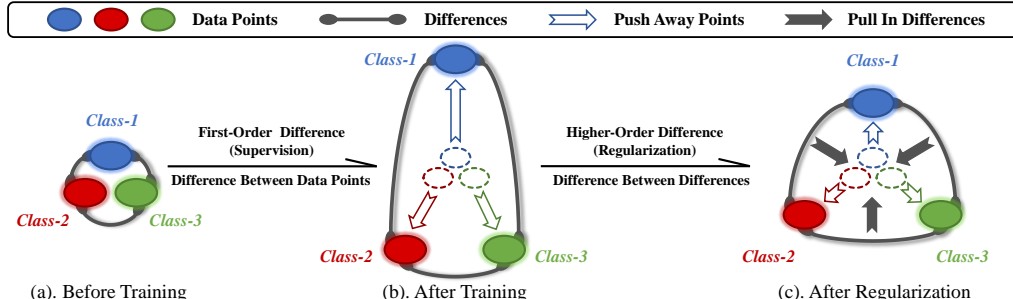

Figure 1: Conceptual illustration of our proposed *difference-aligned regularization* (DAR). Our regularizer minimizes the higher-order difference to find the similar (first-order) representation differences, so that we can use consistent differences for data discrimination.

and positive pairs, respectively. Nevertheless, the above issue of inconsistent distances/differences in supervised metric learning may also exist in self-supervised contrastive learning. The abrupt changes in the distances between different clusters make it difficult for the learning algorithm to capture category information effectively, because the clusters with relatively small mutual distances tend to be misconstrued as a single class. In this case, the discriminability of the pairwise distance is also affected, and thereby reducing the reliability of the learned representation.

For both supervised and unsupervised scenarios, similarity learning has shown promising results, especially in vision and language tasks. However, most existing approaches usually focus on how to constrain the *representation difference* (i.e., the disagreement between the embeddings of two instances) to be consistent with the corresponding similarity supervision (as shown in Fig. 1(a)-(b)). They can hardly ensure the consistency among representation differences (i.e., the difference between differences), and these drastically changed differences may disorder the cluster distributions in the representation space. This will lead to some *overfitting* results where the model prediction is *unstable/unsmooth* w.r.t. the input data, and thus degrading the final classification performance. The popular *adversarial training* (Kim et al., 2020a; Jiang et al., 2020) and classical *regularization techniques* (e.g., the $\ell_1/\ell_2$-*regularizer* (Ying et al., 2009; Zadeh et al., 2016) and *label smoothing* (Wu et al., 2022)) are able to mitigate this impact by enriching the critical training data or constraining the hypotheses. However, they cannot solve this issue in essence to avoid the inconsistency among representation differences. Recent works restricted the local neighbors to having similar distance distributions (Cho et al., 2010; Tian et al., 2019; Wei et al., 2021), but they can only consider the single scale of the distance value, and their constraints are limited to the adjacent relationship.

In this paper, we follow the general principle of *Occam's razor* (Rasmussen & Ghahramani, 2000) to propose a novel *difference alignment regularization* (DAR) that explicitly encourages representation differences to be as close as possible. Specifically, we first construct a *cross-total-variation* (CTV) norm equipped with a generic distance function to measure the disagreement between any two representation differences, and this effectively characterizes the *higher-order difference* information. Such a CTV norm is converted to an equivalent stochastic form whose computational complexity is determined only by the small batch size. We jointly learn the traditional empirical loss with our proposed regularizer to form the *difference-aligned similarity learning* (DASL), so that we can obtain consistent representation differences to distinguish data points from each other (as shown in Fig. 1(c)). Theoretically, we prove that the *stability* and *generalizability* of DASL can be successfully improved with our proposed regularizer. Since DAR is quite a general technique and applicable to many empirical losses, we implement it in both metric learning and contrastive learning approaches, and the baseline results can be significantly improved by our method (e.g., the $2.1\%$ improvement for the self-supervised learning task on *ImageNet-1K*).

Our main contributions are as follows. (i) We propose a novel regularization technique to solve the problem of inconsistent differences with complete theoretical guarantees. (ii) We construct a *strictly defined norm* to measure the higher-order difference, which can be easily optimized with the stochastic gradient. (iii) Experiments demonstrate that our method can outperform the state-of-the-art approaches in both supervised and unsupervised tasks.

## 2 BACKGROUND AND RELATED WORK

As the supervised and unsupervised settings of similarity-based representation learning, the related works of metric learning and contrastive learning are reviewed in this section. Throughout this paper,

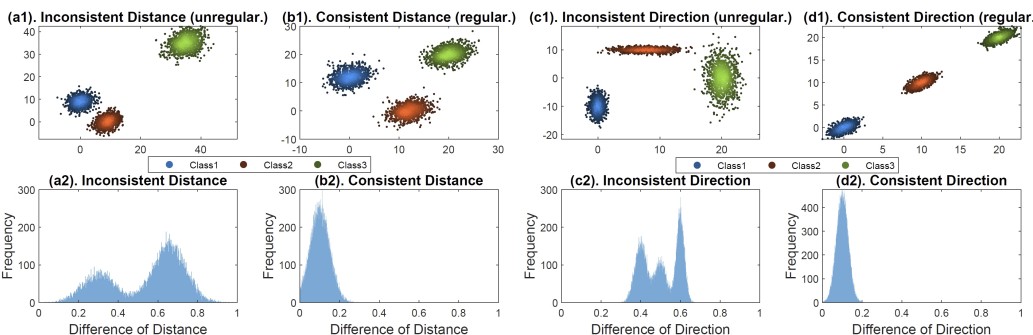

Figure 2: Toy examples to show the consistent/inconsistent differences (including distance values and distance directions). Each column is a data distribution and the corresponding difference distribution of distances (including distance values or distance directions).

we write matrices, vectors, and three-order tensors as bold uppercase characters, bold lowercase characters, and bold calligraphic uppercase characters, respectively. $\|\boldsymbol{x}\|_1$ and $\|\boldsymbol{x}\|_2$ denote the $\ell_1$-norm and $\ell_2$-norm of the vector $\boldsymbol{x}$, respectively.

**Supervised Metric Learning.** Here the pairwise similarities of training data are annotated by humans, and the central problem is how to learn a distance metric or feature representation that faithfully reflects the pairwise similarity between each pair of instances. Both linear (Weinberger et al., 2006; Xu et al., 2018) and nonlinear approaches (Chen et al., 2019; Liao et al., 2023) learn a generic feature representation $\boldsymbol{\varphi} : \mathbb{R}^m \rightarrow \mathbb{R}^h$ (where $h$ is the feature dimensionality), and the corresponding learnable distance is $d_{\boldsymbol{\varphi}}(\boldsymbol{x}, \widehat{\boldsymbol{x}}) = \|\boldsymbol{\varphi}(\boldsymbol{x}) - \boldsymbol{\varphi}(\widehat{\boldsymbol{x}})\|_2$, which measures the similarity between instances $\boldsymbol{x}, \widehat{\boldsymbol{x}} \in \mathbb{R}^m$. The basic learning objective is to reduce the distance $d_{\boldsymbol{\varphi}}(\boldsymbol{x}, \widehat{\boldsymbol{x}})$ if $\boldsymbol{x}$ and $\widehat{\boldsymbol{x}}$ are similar and to enlarge it if $\boldsymbol{x}$ and $\widehat{\boldsymbol{x}}$ are dissimilar. To alleviate the *overfitting* issue caused by the strong nonlinearity of $\boldsymbol{\varphi}$ implemented by deep neural networks, various loss functions (e.g., *Augular loss* (Wang et al., 2017) and *multi-tuplet/Npair loss* (Sohn, 2016)) and metric forms (e.g., *multi-local metrics* (Ye et al., 2016; 2019)) have been proposed to construct the generalized data pair containing three or more data points, which successfully improve the diversity of training data. The regularization technique is another way to reduce overfitting, and recent works have further introduced *adversarial training* (Chen et al., 2018; Duan et al., 2018) or *neighborhood consistency* (Cho et al., 2010; Tian et al., 2019; Wei et al., 2021) to constrain the hypothesis space of learning algorithms.

**Unsupervised (Self-Supervised) Contrastive Learning.** This task has a similar training phase (i.e., considering the pairwise relationship) with metric learning. Existing *noise contrastive estimation* (NCE) loss based methods usually have two critical components: *instance discrimination* to generate *negative pairs* (Wu et al., 2018; Dosovitskiy et al., 2014) and *data augmentation* to generate *positive pairs* (Chen et al., 2020; Jiang et al., 2020). Contrastive learning reduces the distance $d_{\boldsymbol{\varphi}}(\boldsymbol{x}, \boldsymbol{x}^+)$ for a positive pair $(\boldsymbol{x}, \boldsymbol{x}^+)$ and enlarges the distance $d_{\boldsymbol{\varphi}}(\boldsymbol{x}, \boldsymbol{x}_{b_j})$ for negative pairs $\{(\boldsymbol{x}, \boldsymbol{x}_{b_j})\}_{j=1}^n$, where $n$ is the (negative) batch size. Many works further enriched the contrast information within the positive pairs based on different perspectives such as *adversarial perturbation* (Kim et al., 2020a; Jiang et al., 2020) and *multi-view/multi-modal learning* (Tian et al., 2020b; Sun et al., 2022). Since $\boldsymbol{x}$ and $\boldsymbol{x}_{b_j}$ might be semantically similar, yet they are undeservedly pushed away from each other, recent works put forward to correct/reduce the false negative pairs by conventional techniques such as *positive-unlabeled learning* (Chuang et al., 2020) and *pseudo-labeling* (Zheng et al., 2021). Distance regularization approaches have also been proposed to reduce the inconsistency of *distance distributions* (Wei et al., 2021) or to constrain the reliable range of distance values (Chen et al., 2021).

## 3 METHODOLOGY

In this section, we formulate our proposed regularizer and the corresponding learning objective.

### 3.1 MOTIVATION

We start with toy examples to intuitively understand the importance of the consistent differences. In the first case, we generate data points from three *Gaussian distributions*, where the parameters of the three distributions are configured with different sets of means and covariances. Here, the

*inter-cluster* distances are more consistent in Fig. 2(b1) than in Fig. 2(a1). In the second case, we further generate data points from different distributions to form clusters in different directions. The direction distributions in Fig. 2(d1) (i.e., the single direction) are more consistent than those in Fig. 2(c1). For the two cases above, we record the corresponding difference distributions of distance values or distance directions (as shown in Fig. 2(a2)-(d2)). Meanwhile, we sample additional data points from the same distribution to collect the *test data*.

**The Strong Consistency Promotes the Stable Generalizability.** In the above Fig. 2, all the different settings can successfully distinguish clusters from each other, i.e., the *training data* is always discriminated well. However, the inconsistent and consistent cases have different *generalizability*. When the distances/directions are more consistent (i.e., the differences in Fig. 2(b2) and (d2) are close to zero), the clusters become more uniformly distributed, so that the classification reliability on the unseen test data can be improved (see the corresponding test accuracies in Appendix). Our theoretical results also reveal that the consistent distances promote stable generalizability. This is a critical property of a reliable similarity metric, but it is neglected in most existing approaches, so we propose to explicitly characterize this property with a general-purpose regularizer.

## 3.2 FORMULATION

As the above issue of inconsistent difference may exist in both supervised metric learning and unsupervised contrastive learning tasks, here we first show that their loss functions can be simply unified.

**A Unified Form of Supervised and Unsupervised Losses.** For the widely used $(n{+}1)$-tuplet/Npair loss in metric learning and the NCE loss in contrastive learning, based on the given training set $\mathscr{X} = \{\boldsymbol{x}_i \in \mathbb{R}^m | i = 1, 2, \ldots, N\}$ ($m$ is the data dimensionality and $N$ is the sample size), we aim to learn a generic feature representation $\boldsymbol{\varphi} : \mathbb{R}^m \to \mathbb{R}^h$ with the following empirical loss

$$\mathcal{L}_{\text{emp}}(\boldsymbol{\varphi}; \mathscr{X}) = \mathbb{E}_{\boldsymbol{x}, \{b_j\}_{j=1}^n} \left[ -\log \frac{e^{-d_{\boldsymbol{\varphi}}(\boldsymbol{x}, \boldsymbol{x}^+)/\gamma}}{e^{-d_{\boldsymbol{\varphi}}(\boldsymbol{x}, \boldsymbol{x}^+)/\gamma} + \sum_{j=1}^n e^{-d_{\boldsymbol{\varphi}}(\boldsymbol{x}, \boldsymbol{x}_{b_j})/\gamma}} \right], \quad (1)$$

where $\gamma > 0$ is a temperature parameter, and the anchor instance $\boldsymbol{x}$ is sampled from $\mathscr{X}$. Here $\boldsymbol{x}^+$ is directly obtained from the perturbed $\boldsymbol{x}$ for the unsupervised (contrastive learning) case, or it is chosen randomly from the *intra-class* set $\mathscr{X}^+(\boldsymbol{x}) = \{\boldsymbol{z} \,|\, y_{\boldsymbol{z}} = y_{\boldsymbol{x}}, \boldsymbol{z} \in \mathscr{X} \backslash \{\boldsymbol{x}\}\}$ for the supervised (metric learning) case. Correspondingly, the mini-batch instances $\{\boldsymbol{x}_{b_1}, \boldsymbol{x}_{b_2}, \ldots, \boldsymbol{x}_{b_n}\}$ are directly selected from $\mathscr{X} \backslash \{\boldsymbol{x}\}$ for the unsupervised case, or from the *inter-class* set $\mathscr{X}^-(\boldsymbol{x}) = \{\boldsymbol{z} \,|\, y_{\boldsymbol{z}} \neq y_{\boldsymbol{x}}, \boldsymbol{z} \in \mathscr{X}\}$ for the supervised case ($y_{\boldsymbol{x}}$ is the class label of $\boldsymbol{x}$).

Minimizing the above Eq. (1) will constrain the first-order difference $\nabla_{\boldsymbol{\varphi}}^{(1)}(\boldsymbol{x}, \widehat{\boldsymbol{x}}) = \boldsymbol{\varphi}(\boldsymbol{x}) - \boldsymbol{\varphi}(\widehat{\boldsymbol{x}})$ to be consistent with the corresponding (pseudo) supervision of the two instances $\boldsymbol{x}$ and $\widehat{\boldsymbol{x}}$, because the distance value $d_{\boldsymbol{\varphi}}(\boldsymbol{x}, \widehat{\boldsymbol{x}})$ is actually the $\ell_2$-norm of the *first-order* difference $\nabla_{\boldsymbol{\varphi}}^{(1)}(\boldsymbol{x}, \widehat{\boldsymbol{x}})$. Here we want to reduce the inconsistency among these first-order differences (as shown in Fig. 3), so we need to further consider the *higher-order* difference

$$\nabla_{\boldsymbol{\varphi}}^{(2)}(\boldsymbol{x}, \widehat{\boldsymbol{x}}, \boldsymbol{z}, \widehat{\boldsymbol{z}}) = \nabla_{\boldsymbol{\varphi}}^{(1)}(\boldsymbol{x}, \widehat{\boldsymbol{x}}) - \nabla_{\boldsymbol{\varphi}}^{(1)}(\boldsymbol{z}, \widehat{\boldsymbol{z}}), \quad (2)$$

where $(\boldsymbol{x}, \widehat{\boldsymbol{x}})$ and $(\boldsymbol{z}, \widehat{\boldsymbol{z}})$ are two given data pairs, and we use a *vector-valued even function* $\mathcal{G}(\cdot) : \mathbb{R}^h \to \mathbb{R}^+$ to measure such a higher-order

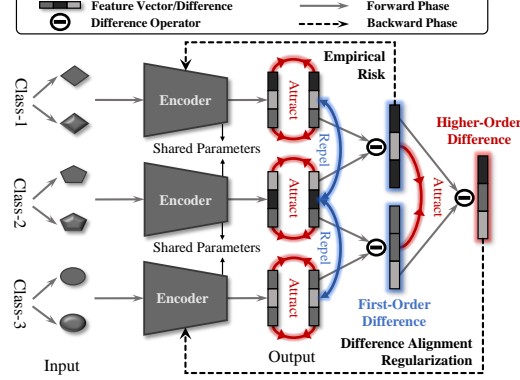

Figure 3: The overall framework of our proposed difference-aligned similarity learning.

difference, namely $\mathcal{G}(\nabla_{\boldsymbol{\varphi}}^{(2)}(\boldsymbol{x}, \widehat{\boldsymbol{x}}, \boldsymbol{z}, \widehat{\boldsymbol{z}}))$. Then we formulate our regularization term in detail.

**Difference Alignment Regularization.** For $N$ training examples of $\mathscr{X}$, we have $\mathrm{C}_N^2$ first-order differences. However, as the intra-class differences are already limited to small values by the empirical loss, we only need to consider the inter-class differences $\{\nabla_{\boldsymbol{\varphi}}^{(1)}(\boldsymbol{x}_i, \boldsymbol{x}_j)\}_{1 \leq i < j \leq N, y_i \neq y_j}$. Correspondingly, when we further consider the higher-order differences, we calculate the following

---

**Algorithm 1** Solving Eq. (8) via SGD.

---

**Input:** training set $\mathscr{X} = \{\boldsymbol{x}_i\}_{i=1}^N$; step size $\eta > 0$; regularization parameter $\lambda > 0$; batch size $n \in \mathbb{N}_+$; randomly initialized $\boldsymbol{\varphi}^{(0)}$; maximum iteration number $T$.

**For** $t$ **from** 1 **to** $T$:

    1). Uniformly pick $(n+1)$ instances $\{\boldsymbol{x}_{b_j}\}_{j=0}^n$ from $\mathscr{X}$;

    2). Compute the gradients of $\mathcal{L}_{\text{emp}}(\boldsymbol{\varphi}; \{\boldsymbol{x}_{b_j}\}_{j=0}^n)$ and $\mathcal{R}_{\text{align}}(\boldsymbol{\varphi}; \{\boldsymbol{x}_{b_j}\}_{j=0}^n)$ (namely $\partial \mathcal{L}_{\text{emp}}(\boldsymbol{\varphi}; \{\boldsymbol{x}_{b_j}\}_{j=0}^n)/\partial \boldsymbol{\varphi}$ and $\partial \mathcal{R}_{\text{align}}(\boldsymbol{\varphi}; \{\boldsymbol{x}_{b_j}\}_{j=0}^n)/\partial \boldsymbol{\varphi}$, respectively);

    3). Update the learning parameter:

$$\boldsymbol{\varphi}^{(t)} = \boldsymbol{\varphi}^{(t-1)} - \eta(\partial \mathcal{L}_{\text{emp}}(\boldsymbol{\varphi}; \{\boldsymbol{x}_{b_j}\}_{j=0}^n)/\partial \boldsymbol{\varphi} + \lambda \partial \mathcal{R}_{\text{align}}(\boldsymbol{\varphi}; \{\boldsymbol{x}_{b_j}\}_{j=0}^n)/\partial \boldsymbol{\varphi}); \quad (7)$$

**End**
**Output:** the converged $\boldsymbol{\varphi}^{(T)}$.

---

cumulative sum of all cases:

$$\sum\nolimits_{1 \le i < j \le N,\, 1 \le k < l \le N,\, (i,j) \ne (k,l), y_i \ne y_j, y_k \ne y_l} \mathcal{G}\left(\nabla_{\boldsymbol{\varphi}}^{(2)}(\boldsymbol{x}_i, \boldsymbol{x}_j, \boldsymbol{x}_k, \boldsymbol{x}_l)\right). \quad (3)$$

We define the *cross-total-variation* (CTV) norm on a matrix $\boldsymbol{M} \in \mathbb{R}^{h \times H}$ ($H \in \mathbb{N}^+$) as

$$\|\boldsymbol{M}\|_{\text{ctv}} = \sum\nolimits_{1 \le j < k \le H} \mathcal{G}(\boldsymbol{M}_j - \boldsymbol{M}_k), \quad (4)$$

where $\boldsymbol{M}_j$ is the $j$-th column of $\boldsymbol{M}$, and we can rewrite Eq. (3) to a value of CTV norm:

$$\sum\nolimits_{1 \le i < j \le N,\, 1 \le k < l \le N,\, (i,j) \ne (k,l), y_i \ne y_j, y_k \ne y_l} \mathcal{G}\left(\nabla_{\boldsymbol{\varphi}}^{(2)}(\boldsymbol{x}_i, \boldsymbol{x}_j, \boldsymbol{x}_k, \boldsymbol{x}_l)\right)$$
$$= 2\left\|\left[\nabla_{\boldsymbol{\varphi}}^{(1)}(\boldsymbol{x}_1, \boldsymbol{x}_2), \ldots, \nabla_{\boldsymbol{\varphi}}^{(1)}(\boldsymbol{x}_i, \boldsymbol{x}_j), \ldots, \nabla_{\boldsymbol{\varphi}}^{(1)}(\boldsymbol{x}_{N-1}, \boldsymbol{x}_N)\right]_{1 \le i < j \le N, y_i \ne y_j}\right\|_{\text{ctv}}. \quad (5)$$

Here the even function $\mathcal{G}(\nabla_{\boldsymbol{\varphi}}^{(2)}(\boldsymbol{x}_i, \boldsymbol{x}_j, \boldsymbol{x}_k, \boldsymbol{x}_l)) = \mathcal{G}(\nabla_{\boldsymbol{\varphi}}^{(2)}(\boldsymbol{x}_k, \boldsymbol{x}_l, \boldsymbol{x}_i, \boldsymbol{x}_j))$ and each element will be twice calculated in Eq. (3), so the corresponding CTV norm has an additional coefficient 2. Furthermore, we have the following theorem to ensure that Eq. (4) is a *strictly defined norm* that satisfies the *non-negativity*, *homogeneity*, and *triangle property* (the proof is given in Appendix).

**Theorem 1.** *The function* $\|\cdot\|_{\text{ctv}}: \mathbb{R}^{h \times H} \to \mathbb{R}^+$ *is a strictly defined norm if and only if the measure function* $\mathcal{G}(\cdot): \mathbb{R}^h \to \mathbb{R}^+$ *is a strictly defined norm.*

In practice, we can use the popular $\ell_1$-norm or $\ell_2$-norm to implement the function $\mathcal{G}$. However, the calculation of Eq. (5) involves $\mathcal{O}(N^2)$ vectors which can be computationally costly for large-scale data. Instead we can consider the stochastic form of the regularizer w.r.t. a mini batch $\{\boldsymbol{x}_{b_0}, \boldsymbol{x}_{b_1}, \ldots, \boldsymbol{x}_{b_n}\}$, where $n$ is the batch size and $\boldsymbol{x}_{b_0}$ denotes the anchor instance $\boldsymbol{x}$ in Eq. (1) for simplicity. Then we define the following *difference alignment regularization* (DAR)

$$\mathcal{R}_{\text{align}}(\boldsymbol{\varphi}; \mathscr{X}) = \mathbb{E}_{\{b_j\}_{j=0}^n}\left\{\left\|\left[\nabla_{\boldsymbol{\varphi}}^{(1)}(\boldsymbol{x}_{b_0}, \boldsymbol{x}_{b_1}), \ldots, \nabla_{\boldsymbol{\varphi}}^{(1)}(\boldsymbol{x}_{b_i}, \boldsymbol{x}_{b_{i+1}}), \ldots, \nabla_{\boldsymbol{\varphi}}^{(1)}(\boldsymbol{x}_{b_{n-1}}, \boldsymbol{x}_{b_n})\right]\right\|_{\text{ctv}}\right\}, \quad (6)$$

and we can easily get that $\mathcal{R}_{\text{align}}(\boldsymbol{\varphi}; \mathscr{X}) = K\|[\nabla_{\boldsymbol{\varphi}}^{(1)}(\boldsymbol{x}_i, \boldsymbol{x}_j)]_{1 \le i < j \le N, y_i \ne y_j}\|_{\text{ctv}}$ where $K > 0$ is *independent* of the train data $\mathscr{X}$ and the feature representation $\boldsymbol{\varphi}$. It implies that minimizing Eq. (5) and Eq. (6) is *mathematically equivalent* (see Appendix for more details).

**Objective & Optimization.** We integrate the above regularizer in Eq. (5) into the conventional empirical loss in Eq. (1), and we obtain the final learning objective of our *difference-aligned similarity learning* (DASL):

$$\min_{\boldsymbol{\varphi} \in \mathcal{H}} \{\mathcal{F}(\boldsymbol{\varphi}) = \mathcal{L}_{\text{emp}}(\boldsymbol{\varphi}; \mathscr{X}) + \lambda \mathcal{R}_{\text{align}}(\boldsymbol{\varphi}; \mathscr{X})\}, \quad (8)$$

where $\lambda > 0$ is a trade-off parameter and $\mathcal{H}$ is the hypothesis space of the learning parameter. Then we are able to optimize our learning objective Eq. (8) in a stochastic way (as shown in Fig. 3). We need to specify the stochastic terms of both $\mathcal{L}_{\text{emp}}(\boldsymbol{\varphi}; \mathscr{X})$ and $\mathcal{R}_{\text{align}}(\boldsymbol{\varphi}; \mathscr{X})$ for a given mini batch. Here the traditional empirical loss $\mathcal{L}_{\text{emp}}(\boldsymbol{\varphi}; \mathscr{X})$ already has its stochastic form $\mathcal{L}_{\text{emp}}(\boldsymbol{\varphi}; \{\boldsymbol{x}_{b_j}\}_{j=0}^n)$,

and our DAR also has its stochastic form $\mathcal{R}_{\text{align}}(\boldsymbol{\varphi}; \{\boldsymbol{x}_{b_j}\}_{j=0}^n)$ of which the computational complexity only depends on the batch size [1]. Therefore, in each iteration, we can use the stochastic gradient instead of the full gradient for fast computation (see the running time comparison in Appendix).

The detailed iteration steps based on *stochastic gradient descent* (SGD) (Reddi et al., 2016) are summarized in Algorithm 1, and we have the corresponding theoretical result (in Appendix) to investigate the convergence behavior of iteration points $\boldsymbol{\varphi}^{(1)}, \boldsymbol{\varphi}^{(2)}, \ldots, \boldsymbol{\varphi}^{(T)}$ obtained by Eq. (7).

## 4 THEORETICAL ANALYSES

In this section, we theoretically investigate the robustness and generalizability of our method.

**Model Robustness.** For a given data pair $(\boldsymbol{x}, \widehat{\boldsymbol{x}})$, our model prediction is the distance value $d_{\boldsymbol{\varphi}}(\boldsymbol{x}, \widehat{\boldsymbol{x}})$, which measures the similarity between $\boldsymbol{x}$ and $\widehat{\boldsymbol{x}}$. Therefore, we would like to analyze the stability of such a distance function learned from DASL in Eq. (8) to investigate the robustness of our method. Intuitively, for two given data pair $(\boldsymbol{x}, \widehat{\boldsymbol{x}})$ and $(\boldsymbol{z}, \widehat{\boldsymbol{z}})$, our regularizer $\mathcal{R}_{\text{align}}$ has already reduced the higher-order difference, so the inconsistency between the first-order difference $\nabla_{\boldsymbol{\varphi}}^{(1)}(\boldsymbol{x}, \widehat{\boldsymbol{x}})$ and $\nabla_{\boldsymbol{\varphi}}^{(1)}(\boldsymbol{z}, \widehat{\boldsymbol{z}})$ (as well as the divergence between $d_{\boldsymbol{\varphi}}(\boldsymbol{x}, \widehat{\boldsymbol{x}})$ and $d_{\boldsymbol{\varphi}}(\boldsymbol{z}, \widehat{\boldsymbol{z}})$) should be bounded. To be more rigorous, we have the following theorem to reveal such an upper bound.

**Theorem 2.** *Suppose that the instances $\boldsymbol{x}$, $\widehat{\boldsymbol{x}}$, $\boldsymbol{z}$, and $\widehat{\boldsymbol{z}}$ are independently sampled from the same distribution as the training set $\mathscr{X}$. Then, for any feature representation $\boldsymbol{\varphi}$ learned from the objective $\mathcal{F}(\boldsymbol{\varphi})$, we have that with probability at least $1 - \delta$,*

$$|d_{\boldsymbol{\varphi}}(\boldsymbol{x}, \widehat{\boldsymbol{x}}) - d_{\boldsymbol{\varphi}}(\boldsymbol{z}, \widehat{\boldsymbol{z}})| \leq \xi(\lambda)(\|\boldsymbol{x} - \widehat{\boldsymbol{x}}\|_2 + \|\boldsymbol{z} - \widehat{\boldsymbol{z}}\|_2) \max\{d_{\boldsymbol{\varphi}}(\boldsymbol{t}, \widehat{\boldsymbol{t}})|\boldsymbol{t}, \widehat{\boldsymbol{t}} \in \mathscr{X}\} \sqrt{[\ln(2/\delta)]/(2N)},$$
(9)

*where $\xi(\lambda) = L \frac{\mathcal{L}_{emp}(\boldsymbol{\varphi}^{(0)}; \mathscr{X})}{\lambda}$ is monotonically decreasing w.r.t. the regularization parameter $\lambda$, and the constant $L > 0$ is independent of $\boldsymbol{\varphi}$ and $\mathscr{X}$.*

From the above Eq. (9), we can clearly observe that the distance error bound is first affected by the original distances $\|\boldsymbol{x} - \widehat{\boldsymbol{x}}\|_2$ and $\|\boldsymbol{z} - \widehat{\boldsymbol{z}}\|_2$, and it gradually decreases with the increase of the sample size $N$. This is consistent with our intuitions and empirical knowledge. Meanwhile, with the proposed regularizer, such a bound can be further reduced, and the increased regularization parameter will shrink the bound and make the prediction more stable/robust w.r.t. the input data.

**Generalization Error.** We would like to further prove that our learning algorithm also tightens the *generalization error bound* (GEB) (Chen et al., 2021) of the conventional similarity learning approach. As we know, GEB usually has a convergence rate of $\mathcal{O}(1/\sqrt{N})$ for an empirical risk minimization model. Here we do not investigate the convergence rate as a function of sample size, but we show that a tightened GEB benefits from the regularization $\mathcal{R}_{\text{align}}$ to validate the effectiveness of our method. Specifically, for the underlying data distribution $\mathscr{D}$, we denote the expected risk $\widetilde{\mathcal{L}}(\boldsymbol{\varphi}; \mathscr{D}) = \mathbb{E}_{\{\boldsymbol{t}_i | \boldsymbol{t}_i \sim \mathscr{D}\}_{i=1}^N}[\mathcal{L}(\boldsymbol{\varphi}; \{\boldsymbol{t}_i\}_{i=1}^N)]$ and discuss how far it is from the empirical risk $\mathcal{L}(\boldsymbol{\varphi})$.

**Theorem 3.** *For any $\boldsymbol{\varphi}$ learned from the objective $\mathcal{F}(\boldsymbol{\varphi})$ and any given constant $\delta \in (0, 1)$, we have that with probability at least $1 - \delta$,*

$$|\mathcal{L}(\boldsymbol{\varphi}) - \widetilde{\mathcal{L}}(\boldsymbol{\varphi}; \mathscr{D})| \leq \beta(\lambda)\omega(n)\log(1 + \max\{d_{\boldsymbol{\varphi}}(\boldsymbol{t}, \widehat{\boldsymbol{t}})|\boldsymbol{t}, \widehat{\boldsymbol{t}} \in \mathscr{X}\}) \sqrt{[\ln(2/\delta)]/(2N)},$$
(10)

*where $\beta(\lambda) = (C + 2/C)/\lambda$ is monotonically decreasing w.r.t. $\lambda$ and $\omega(n) = \log\left(\frac{e^2}{n} + 1\right)$ is monotonically decreasing w.r.t. $n$. Here the constant $C > 0$ is independent of $\boldsymbol{\varphi}$ and $\mathscr{X}$.*

From the above result in Eq. (10), it is easy to observe that the error bound is dominated by two main aspects. First, the error bound gradually decreases with the increase of the sampling number $N$ as well as the batch size $n$. This is consistent with the observations in existing works (Saunshi et al., 2019). More importantly, we can find that such an error bound becomes *tighter* as $\lambda$ increases, and thus the regularization term $\mathcal{R}_{\text{align}}$ can accelerate the empirical risk convergence to the expected risk. Therefore, Theorem 3 shows that our method successfully improves the generalization ability of conventional similarity learning algorithms.

---

[1]Specifically, $\mathcal{L}_{\text{emp}}(\boldsymbol{\varphi}; \{\boldsymbol{x}_{b_j}\}_{j=0}^n) = \log[e^{-d_{\boldsymbol{\varphi}}(\boldsymbol{x}_{b_0}, \boldsymbol{x}_{b_0}^+)/\gamma}/(e^{-d_{\boldsymbol{\varphi}}(\boldsymbol{x}_{b_0}, \boldsymbol{x}_{b_0}^+)/\gamma} + \sum_{j=1}^n e^{-d_{\boldsymbol{\varphi}}(\boldsymbol{x}_{b_0}, \boldsymbol{x}_{b_j}^+)/\gamma})]$ and $\mathcal{R}_{\text{align}}(\boldsymbol{\varphi}; \{\boldsymbol{x}_{b_j}\}_{j=0}^n) = \|[\nabla_{\boldsymbol{\varphi}}^{(1)}(\boldsymbol{x}_{b_0}, \boldsymbol{x}_{b_1}), \ldots, \nabla_{\boldsymbol{\varphi}}^{(1)}(\boldsymbol{x}_{b_0}, \boldsymbol{x}_{b_j}), \ldots, \nabla_{\boldsymbol{\varphi}}^{(1)}(\boldsymbol{x}_{b_0}, \boldsymbol{x}_{b_n})]\|_{\text{ctv}}$.

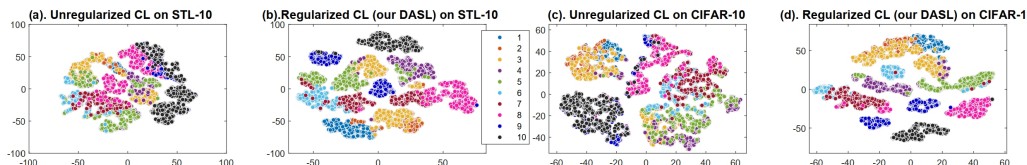

Figure 4: The t-SNE visualizations of our DASL and the conventional (unregularized) contrastive learning (CL) method on STL-10 and CIFAR-10 datasets, where we can clearly observe that our method obtains the better clustering results than the unregularized method.

## 5 EXPERIMENTAL RESULTS

In this section, we show experimental results on real-world datasets to validate the effectiveness of our DASL in both the supervised metric learning and unsupervised contrastive learning tasks. We first provide ablation studies and visualization results. Then, we compare our method with existing state-of-the-art methods. Both the training and test processes are implemented on Pytorch (Paszke et al., 2019) with TeslaV100 GPUs, where the regularization parameter $\lambda$ is set to $0.5$. The dimensionality $h$ and the parameter $\gamma$ in Eq. (1) are set to $512$ and $0.2$, respectively. The hyper-parameters of compared methods are set to the recommended values according to their original papers.

### 5.1 ABLATION STUDIES & VISUALIZATION RESULTS

In this subsection, we would like to first investigate the usefulness of our proposed regularizer in both supervised and unsupervised scenarios. Here we use the typical $\ell_2$-norm (considering its smoothness in the optimization) to implement the measure function $\mathcal{G}(\cdot)$ in our regularizer.

**Supervised Metric Learning.** We employ different backbones (*BN-Inception* (Ioffe & Szegedy, 2015) for *Npair* (Sohn, 2016), and *ResNet-50* (He et al., 2016) for *ProxyAnchor* (Kim et al., 2020b) and *MetricFormer* (Yan et al., 2022)) to validate the effectiveness of our approach in various settings. We conduct experiments on *CAR* (Krause et al., 2013) and *CUB* (Welinder et al., 2010) datasets and record the test accuracy of compared methods (with 500 epochs, learning rate = $10^{-3}$, and

Table 1: Classification accuracy rates of baseline methods and our method on CAR and CUB datasets (feature embedding sizes are 128 and 512).

| METHOD | CAR | | | | CUB | | | |
|---|---|---|---|---|---|---|---|---|
| | 128-dim. | | 512-dim. | | 128-dim. | | 512-dim. | |
| | R@1 | R@8 | R@1 | R@8 | R@1 | R@8 | R@1 | R@8 |
| Npair(BN) w/o Reg. ($\lambda = 0$) | 68.36 | 86.01 | 82.37 | 95.12 | 58.12 | 78.72 | 65.38 | 90.82 |
| DASL[Npair(BN)+($\lambda = 0.1$)] | 68.36 | 88.28↑ | 85.30↑ | 96.10↑ | 58.12 | 80.35↑ | 66.33↑ | 91.89↑ |
| DASL[Npair(BN)+($\lambda = 0.5$)] | **70.28↑** | **90.36↑** | **89.22↑** | **96.30↑** | **62.02↑** | **82.30↑** | **69.12↑** | **92.53↑** |
| ProxA.(R50) w/o Reg. ($\lambda = 0$) | 69.24 | 87.86 | 87.71 | 97.86 | 62.12 | 79.26 | 69.72 | 92.41 |
| DASL[ProxA.+($\lambda = 0.1$)] | 69.24 | 88.86↑ | 89.86↑ | 98.23↑ | 62.12 | 80.95↑ | 71.13↑ | 92.78↑ |
| DASL[ProxA.+($\lambda = 0.5$)] | **70.28↑** | **91.21↑** | **92.31↑** | **98.90↑** | **63.13↑** | **82.17↑** | **73.96↑** | **94.21↑** |
| M.F.(R50) w/o Reg ($\lambda = 0$) | 72.42 | 89.53 | 91.76 | 97.21 | 69.33 | 85.12 | 74.42 | 92.53 |
| DASL[MetricF.+($\lambda = 0.1$)] | **73.56↑** | 91.12↑ | 91.56 | 97.53↑ | 70.86↑ | 86.25↑ | 74.55↑ | 92.97↑ |
| DASL[MetricF.+($\lambda = 0.5$)] | **73.56↑** | **92.32↑** | **92.12↑** | **98.41↑** | **72.35↑** | **88.28↑** | **75.51↑** | **93.31↑** |

batch size = 512 (Zhou et al., 2021; 2022a)) in Tab. 1. We can observe that our method can work well with all the three baseline methods. Our DASL obtains relatively stable performance in various cases with different embedding sizes. When we increase the regularization parameter $\lambda$ from $0.1$ to $0.5$, the corresponding recognition accuracy rates are also improved, which clearly validates the significance of our regularization term.

**Unsupervised Contrastive Learning.** Here we only adopt the ResNet-50 backbone for all compared methods. In Fig. 5, we show the classification accuracy rates of all compared methods on *CIFAR-10* (Krizhevsky et al., 2009) and *STL-10* datasets (Coates et al., 2011), where we can observe that our method consistently improves the corresponding baseline results (*SimCLR* (Chen et al., 2020), *HCL* (Robinson et al., 2021), and *BYOL* (Grill et al., 2020)) in all cases. In order to have a more intuitive understanding of the regularizer's effect, we conduct the *t-SNE* embedding (Van der Maaten & Hinton, 2008) to obtain the 2-dimensional data points which help us better understand the usefulness of our introduced new term. As shown in Fig. 4, DASL can successfully obtain more accurate separations than the baseline results.

### 5.2 EXPERIMENTS ON SUPERVISED METRIC LEARNING

**Image Retrieval.** Here we investigate the capability of DASL with the popular retrieval tasks on *CAR* (Krause et al., 2013), *CUB* (Welinder et al., 2010), and *SOP* (Oh Song et al., 2016). The further compared methods are *JDR* (Chu et al., 2020), *IBC* (Seidenschwarz et al., 2021),

Figure 5: Classification accuracy of all methods on STL-10 and CIFAR-10 datasets, where the (negative) batch size is from 32 to 512.

*AVSL* (Zhang et al., 2022), and *ContextSimilarity* (Liao et al., 2023). In our experiments, all compared methods are incorporated into the ResNet-50 backbone to train end-to-end network as the final distance metrics. We refer to the combinations of our method with Npair loss and ProxyAnchor loss as DASL-NP and DASL-PA, respectively. The NMI and Recall@R scores of all compared methods are shown in Tab. 2. From the quantitative results, we clearly observe that MetricFormer, ContextSimilarity, and our methods obtain the higher accuracies than other comapred methods. Compared with the two strong baseline methods, our DASL can achieve either better or competitive NMI and Recall@R scores on the three datasets.

Table 2: Performance of all compared methods (with ResNet-50 backbone) on *CAR-196, CUB-200, and SOP* datasets. The best two results are **bolded** and underlined, respectively.

| METHOD | CAR-196 | | | | CUB-200 | | | | SOP | | | |
|---|---|---|---|---|---|---|---|---|---|---|---|---|
| | NMI | R@1 | R@4 | R@8 | NMI | R@1 | R@4 | R@8 | NMI | R@1 | R@10 | R@100 |
| Npair(Sohn, 2016) | 69.50 | 82.57 | 94.97 | 95.92 | 69.53 | 64.52 | 85.63 | 91.15 | 91.11 | 76.21 | 88.43 | 92.08 |
| ProxyA.(Kim et al., 2020b) | 75.72 | 87.71 | 95.76 | 97.86 | 72.31 | 69.72 | 87.01 | 92.41 | 91.02 | 78.39 | 90.48 | 96.16 |
| JDR(Chu et al., 2020) | 70.56 | 84.86 | 94.56 | 97.21 | 70.32 | 69.44 | 87.01 | 91.33 | 92.21 | 79.21 | 90.53 | 96.01 |
| IBC(Seidenschwarz et al., 2021) | 74.82 | 88.11 | 96.21 | 98.21 | 74.01 | 70.32 | 87.61 | 92.72 | 92.61 | 81.42 | 91.32 | 95.89 |
| AVSL(Zhang et al., 2022) | 75.86 | 91.51 | 97.02 | 98.41 | 73.21 | 71.91 | 88.11 | 93.21 | 91.21 | 79.61 | 91.40 | 96.40 |
| MetricF.(Yan et al., 2022) | 76.23 | 91.76 | 96.31 | 97.21 | 75.41 | **74.42** | 85.75 | 92.53 | 92.71 | 82.23 | 92.62 | 96.33 |
| ContextS.(Liao et al., 2023) | 76.32 | 91.80 | 97.14 | 98.41 | 74.01 | 71.91 | 88.82 | 93.42 | 92.61 | 82.63 | 92.56 | 96.74 |
| DASL-NP (ours) | 75.96↑ | 86.34↑ | 97.56↑ | 98.87↑ | 73.52↑ | 69.63↑ | 89.62↑ | 93.61↑ | 92.85↑ | 79.21↑ | 93.21↑ | 97.86↑ |
| DASL-PA (ours) | **77.32**↑ | **92.31**↑ | **97.82**↑ | **98.90**↑ | **76.50**↑ | 73.96↑ | **90.54**↑ | **94.21**↑ | **93.86**↑ | **83.32**↑ | **93.86**↑ | **97.95**↑ |

**Face Recognition.** Here *CASIA-WebFace* (Yi et al., 2014) is employed as the training set while the test sets include *AgeDB30* (Moschoglou et al., 2017), *CFP-FP* (Sengupta et al., 2016), and *MegaFace* (Kemelmacher-Shlizerman et al., 2016). The batch size is set to 256 and embedding size to 512 for all methods (with ResNet-50 backbone). The compared methods are dif-

Table 3: Accuracy rates of all compared methods on AgeDB30, CFPFP, and MegaFace datasets.

| METHOD | Face Verification | | Face Identification (MegaFace) | | |
|---|---|---|---|---|---|
| | Age. | CFP. | Mega-$10^6$ | Mega-$10^5$ | Mega-$10^4$ |
| Softmax | 91.30 | 93.39 | 80.43 | 87.11 | 92.83 |
| Sph.+$\ell_2$-Reg(Liu et al., 2017) | 93.42 | 94.30 | 88.38 | 92.86 | 95.93 |
| Sph.+SEC(Zhang et al., 2020) | 93.45 | 94.39 | 88.42 | 92.79 | 95.88 |
| Arc.+$\ell_2$-Reg(Deng et al., 2019) | 93.93 | 94.77 | 90.68 | 94.34 | 96.83 |
| Arc.+SEC(Zhang et al., 2020) | 93.82 | 94.91 | 90.91 | 94.56 | 96.95 |
| DASL (Sph.+DAR) | 94.21↑ | 95.21↑ | 89.86↑ | 93.96↑ | 96.25↑ |
| DASL (Arc.+DAR) | **95.33**↑ | **96.15**↑ | **91.35**↑ | **95.76**↑ | **97.28**↑ |

ferent regularized versions of Sphereface (Zhang et al., 2020) and Arcface (Deng et al., 2019). As shown in Tab. 3, the two cosine similarity based approaches can always perform better than the original softmax. We also observe that our regularizer improves the performance of both Sphereface and Arcface in all cases. For example, on MegaFace with $10^6$ distractors, the accuracies of Sphereface and Arcface are boosted by $1.44\%$ and $1.20\%$, respectively.

## 5.3 EXPERIMENTS ON UNSUPERVISED CONTRASTIVE LEARNING

**Image Classification.** We employ ResNet-50 as our backbone and implement our method based on *SimCLR* (Chen et al., 2020) and *SwAV* (Caron et al., 2020), and the corresponding results are denoted as DASL (cluster-free) and DASL (cluster-used) respectively. We train our method on *ImageNet-100* and *ImageNet-1K* (Russakovsky et al., 2015), and compare it with existing representative approaches including *contrastive multiview coding* (CMC) (Tian et al., 2020a), *hard negative based contrastive learning* (HCL) (Robinson et al., 2021), *prototypical contrastive learning* (PCL) (Li et al., 2021), *BYOL* (Grill et al., 2020), and *meta augmentation* (MetAug) (Li et al., 2022). Then we also implement our method on the popular *ViT-B/16* backbone and compare it with three more methods including *DINO* (Caron et al., 2021), *iBOT* (Zhou et al., 2022b), and *PQCL* (Zhang et al., 2023). We conduct comprehensive evaluations by recording the classification accuracy rates of all methods obtained with the fine-tuning linear softmax (i.e., the Top-1 score and Top-5 score of *linear probing*) and the $k$-NN classification (here $k = 8$). From Tab. 4, we can observe that our method successfully improves the SimCLR and HCL by at least $4\%$ in different cases of batch size on the two datasets.

Table 4: Classification accuracy (%) of all methods on ImageNet-100 and ImageNet-1K datasets. The batch sizes are set to 1024 and 512 for ResNet-50 and ViT-B/16 backbones, respectively. Here the best and second-best results are **bolded** and underlined, respectively.

| METHOD | ImageNet-100 | | | | | | ImageNet-1K | | | | | | #Arch. |
|---|---|---|---|---|---|---|---|---|---|---|---|---|---|
| | 100 epochs | | | 400 epochs | | | 300 epochs | | | 800 epochs | | | |
| | $k$-NN | Top-1 | Top-5 | $k$-NN | Top-1 | Top-5 | $k$-NN | Top-1 | Top-5 | $k$-NN | Top-1 | Top-5 | |
| SimCLR(Chen et al., 2020) | 55.9 | 61.3 | 78.6 | 70.6 | 75.2 | 92.1 | 64.2 | 67.4 | 87.9 | 66.1 | 69.3 | 89.6 | Res.50 |
| BYOL(Grill et al., 2020) | 56.3 | 65.5 | 77.8 | 69.2 | 73.2 | 90.1 | 66.9 | 71.2 | 90.5 | 67.2 | 73.2 | 91.5 | Res.50 |
| CMC(Tian et al., 2020a) | 57.7 | 60.2 | 79.2 | 71.6 | 73.6 | 92.1 | 63.2 | 68.2 | 87.2 | 67.2 | 71.2 | 89.9 | Res.50 |
| PCL(Li et al., 2021) | 55.9 | 60.2 | 77.2 | 71.5 | 76.1 | 93.2 | 59.5 | 66.5 | 86.7 | 62.2 | 70.5 | 90.5 | Res.50 |
| SwAV(Caron et al., 2020) | 58.2 | 61.0 | 79.4 | 72.1 | 75.8 | 92.9 | 65.4 | 73.1 | 91.2 | 65.7 | 75.3 | 91.5 | Res.50 |
| HCL(Robinson et al., 2021) | 55.9 | 60.8 | 79.3 | 70.2 | 74.6 | 92.3 | 64.2 | 71.2 | 91.2 | 67.2 | 71.7 | 90.7 | Res.50 |
| MetAug(Li et al., 2022) | 59.2 | 61.1 | 79.4 | 69.8 | 75.6 | 93.2 | 65.4 | 74.2 | 91.1 | 67.8 | 76.0 | 92.9 | Res.50 |
| DASL (cluster-free) | 60.5 | 65.2 | 79.8 | 73.5 | 76.6 | 93.9 | 68.3 | 72.7 | 91.9 | 68.2 | **76.7** | 92.9 | Res.50 |
| DASL (cluster-used) | **61.5** | **67.3** | **80.1** | **74.2** | **77.5** | **94.5** | **69.1** | **74.8** | **92.4** | **69.1** | 76.6 | **93.2** | Res.50 |
| BYOL(Grill et al., 2020) | 57.2 | 62.8 | 77.9 | 72.1 | 76.9 | 93.8 | 66.6 | 71.4 | 91.2 | 68.2 | 74.2 | 92.8 | ViT-B/16 |
| SwAV(Caron et al., 2020) | 60.1 | 62.5 | 80.5 | 74.2 | 77.8 | 94.2 | 64.7 | 71.8 | 91.1 | 69.2 | 75.6 | 91.8 | ViT-B/16 |
| DINO(Caron et al., 2021) | 61.5 | 67.5 | 81.8 | 78.2 | 79.2 | 95.5 | 72.3 | 76.1 | 92.4 | 76.2 | 78.2 | 94.2 | ViT-B/16 |
| iBOT(Zhou et al., 2022b) | 61.5 | 68.2 | 82.2 | 77.5 | 78.5 | 95.2 | 71.5 | 75.0 | 91.9 | 75.2 | 76.0 | 92.6 | ViT-B/16 |
| PQCL(Zhang et al., 2023) | 62.3 | 66.7 | 82.5 | 78.5 | 79.5 | 94.8 | 70.8 | 76.5 | 91.9 | **78.3** | 76.9 | 93.0 | ViT-B/16 |
| DASL (cluster-free) | 62.5 | 69.5 | 81.1 | 78.4 | 80.1 | 96.1 | 71.5 | **77.8** | 92.7 | 76.2 | 79.2 | 94.5 | ViT-B/16 |
| DASL (cluster-used) | **63.4** | **69.7** | **82.8** | **79.3** | **82.3** | **96.8** | **72.9** | 76.8 | **93.5** | 77.9 | **79.9** | **96.3** | ViT-B/16 |

Based on the powerful ViT-B/16 encoder, our method also consistently boosts the baseline methods and outperforms the three state-of-the-art methods (DINO, iBOT, and PQCL) in most cases.

**Sentence Embedding.** For the *BookCorpus* dataset which includes six sub-tasks *MR*, *CR*, *SUBJ*, *MPQA*, *TREC*, and *MSRP* (Kiros et al., 2015), we follow the experimental settings in the baseline method *quick-thought* (QT) (Logeswaran & Lee, 2018) to choose the neighboring sentences as positive pairs. Then, we further compare our DASL with *consistent contrast* (CO2) (Wei et al., 2021), and *uncertainty representativeness mixing* (UnReMix) (Tabassum et al., 2022), and the corresponding average classification accuracy rates are shown in Fig. 6. Our method improves the classification accuracy of baseline method QT by at least 2% on most classification benchmarks. This clearly demonstrates that the consistent difference is not only useful in the image data but also in the text data, and our method is a good solution to utilize such a property for model training.

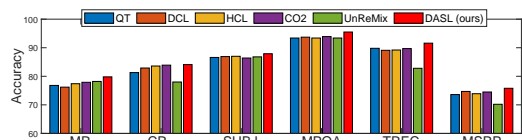

Figure 6: Classification accuracy rates (%) of all compared methods on the BookCorpus dataset including six text classification tasks.

For the *STS* dataset (Agirre et al., 2016), we follow the common practice in *SimCSE* (Gao et al., 2021) to use the pre-trained checkpoints of *BERT* (Devlin et al., 2018), and we further compare our method with *information minimization contrastive learning* (InforMin-CL) (Chen et al., 2022), *misCSE* (Klein & Nabi, 2022), and *smoothed contrastive learning* (SCL) (Wu et al., 2022). As we can observe from Tab. 5, DASL obtains considerable improvements on the baseline method SimCSE. Meanwhile, our method can outperform the other two representative methods misCSE and InforMin-CL in most cases. Our DASL also achieves the best average score in all compared methods.

Table 5: Classification accuracy rates (%) of all compared methods on the STS dataset including five tasks and the corresponding average scores.

| METHOD | STS12 | STS13 | STS14 | STS15 | STS16 | Aver. |
|---|---|---|---|---|---|---|
| SimCSE | 68.69 | 82.05 | 72.91 | 81.15 | 79.39 | 76.84 |
| PCL | 72.74 | 83.36 | 76.05 | 83.07 | 79.26 | 78.90 |
| Inf.Min | 70.22 | 83.48 | 75.51 | 81.72 | 79.88 | 78.16 |
| miCSE | 71.71 | 83.09 | 75.46 | 83.13 | 80.22 | 78.72 |
| SCL | **72.86** | **84.91** | 76.79 | 84.35 | 81.74 | 80.13 |
| SACL (ours) | 72.70 | 84.53 | **78.32** | **85.55** | **82.54** | **80.73** |

## 6    CONCLUSION

In this paper, we investigated the issue of inconsistent representation differences in traditional similarity learning. We proposed a novel DAR regularizer to encourage the learning algorithm to seek for the close differences for data discrimination. To this end, we built a new CTV norm to measure the divergence among representation differences, and we showed that it can be easily solved by SGD. Our proposed DAR is quite a generic technique and we can easily deploy it in both metric learning and contrastive learning tasks with negligible computation overhead. We conducted extensive theoretical analyses that guarantee the effectiveness of our method. Comparison experiments on real-world datasets across multiple domains indicated that our learning algorithm acquires more reliable feature representations than state-of-the-art methods. As this paper merely focused on the unsupervised and fully supervised cases, in the future we plan to explore the inconsistent representation differences in semi-supervised and weakly supervised scenarios.

ACKNOWLEDGMENT

M.S. was supported by JST CREST Grant Number JPMJCR18A2 and a grant from Apple, Inc. Any views, opinions, findings, and conclusions or recommendations expressed in this material are those of the authors and should not be interpreted as reflecting the views, policies or position, either expressed or implied, of Apple Inc. J.Y. was supported by the National Science Fund of China under Grant No. 62361166670.

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

APPENDIX

This part is the appendix of our manuscript. It includes the additional experiments and the mathematical proofs of theorems.

## A ADDITIONAL EXPERIMENTS

### A.1 NUMERICAL RESULTS OF FIG. 2

In Fig. 2, the 1000 points of each cluster (as the training data) are generated from the *Gaussian distributions*, and the corresponding distribution parameters are listed in Tab. 6. Meanwhile, we sample additional data points from the same distribution to collect the test data (200 points for each cluster).

We conduct $k$-NN on both training and test data to record the corresponding classification accuracy rates. The results are listed in Tab. 7, where we can clearly find that all cases have achieved the satisfied training accuracy. However, the cases of consistent differences can obtain better test results than the cases of inconsistent differences.

Table 6: The distribution parameters (the mean vector and covariance matrix) of clusters in Fig. 2.

| | (a1), inconsistent | (b1), consistent | (c1), inconsistent | (d1), consistent |
|---|---|---|---|---|
| Cluster1 | $\mathcal{N}\left(\begin{bmatrix}0\\9\end{bmatrix}, \begin{bmatrix}4&1\\1&4\end{bmatrix}\right)$ | $\mathcal{N}\left(\begin{bmatrix}0\\12\end{bmatrix}, \begin{bmatrix}4&1\\1&4\end{bmatrix}\right)$ | $\mathcal{N}\left(\begin{bmatrix}0\\-10\end{bmatrix}, \begin{bmatrix}1&0\\0&6\end{bmatrix}\right)$ | $\mathcal{N}\left(\begin{bmatrix}0\\0\end{bmatrix}, \begin{bmatrix}1&0.75\\0.75&1\end{bmatrix}\right)$ |
| Cluster2 | $\mathcal{N}\left(\begin{bmatrix}9\\0\end{bmatrix}, \begin{bmatrix}4&1\\1&4\end{bmatrix}\right)$ | $\mathcal{N}\left(\begin{bmatrix}12\\0\end{bmatrix}, \begin{bmatrix}4&1\\1&4\end{bmatrix}\right)$ | $\mathcal{N}\left(\begin{bmatrix}8\\10\end{bmatrix}, \begin{bmatrix}6&0\\0&1\end{bmatrix}\right)$ | $\mathcal{N}\left(\begin{bmatrix}10\\10\end{bmatrix}, \begin{bmatrix}1&0.75\\0.75&1\end{bmatrix}\right)$ |
| Cluster3 | $\mathcal{N}\left(\begin{bmatrix}35\\35\end{bmatrix}, \begin{bmatrix}4&1\\1&4\end{bmatrix}\right)$ | $\mathcal{N}\left(\begin{bmatrix}20\\20\end{bmatrix}, \begin{bmatrix}4&1\\1&4\end{bmatrix}\right)$ | $\mathcal{N}\left(\begin{bmatrix}20\\0\end{bmatrix}, \begin{bmatrix}2&0\\0&24\end{bmatrix}\right)$ | $\mathcal{N}\left(\begin{bmatrix}20\\20\end{bmatrix}, \begin{bmatrix}1&0.75\\0.75&1\end{bmatrix}\right)$ |

Table 7: Training and test accuracy rates of all cases (%).

| | (a1), Training/Test | (b1), Training/Test | (c1), Training/Test | (d1), Training/Test |
|---|---|---|---|---|
| Cluster1 | 99.1/75.3 | 99.1/85.6 | 98.5/85.1 | 98.5/85.1 |
| Cluster2 | 99.2/76.3 | 98.5/86.7 | 99.1/85.2 | 98.5/86.4 |
| Cluster3 | 99.8/89.3 | 98.8/95.3 | 98.1/74.1 | 98.1/85.2 |
| Average | 99.4/80.3 | 98.8/89.2 | 98.6/81.5 | 98.4/85.6 |

### A.2 EXPERIMENTS ON GRAPH EMBEDDING

We test our method on challenging graph data to further investigate the generalizability of our method. Here two types of datasets are considered, those are the biochemical molecules and the social networks, including *DD*, *PTC*, *IMDB-B*, *IMDB-M*, *RDT-B*, *PROTEINS*, *NCI1*, and *MUTAG* (Yanardag & Vishwanathan, 2015).

Here we use the representative method *InfoGraph* (Sun et al., 2020) as the baseline, and we follow the common practice for the downstream graph-level classification task on datasets. Note here we fine-tune a *support vector machine* (SVM) (Cortes & Vapnik, 1995) on the learned feature representation to evaluate the final classification performance by using the 10-fold cross-validation. We split the dataset into the training, test, and validation sets at the proportion of $8/1/1$ and report the mean classification accuracy with standard deviation after 5 runs followed by a linear SVM classifier. The SVM is trained using cross-validation on training folds of data and the model for testing is selected by the best validation performance. Our compared methods include HCL, *GraphCL* (You et al., 2020), *JOAO* (You et al., 2021), and *GroupCL* (Xu et al., 2022).

From the results shown in Fig. 7, we observe that our DASL consistently improves the baseline method InfoGraph in all eight cases. Meanwhile, compared with the other graph contrastive learning

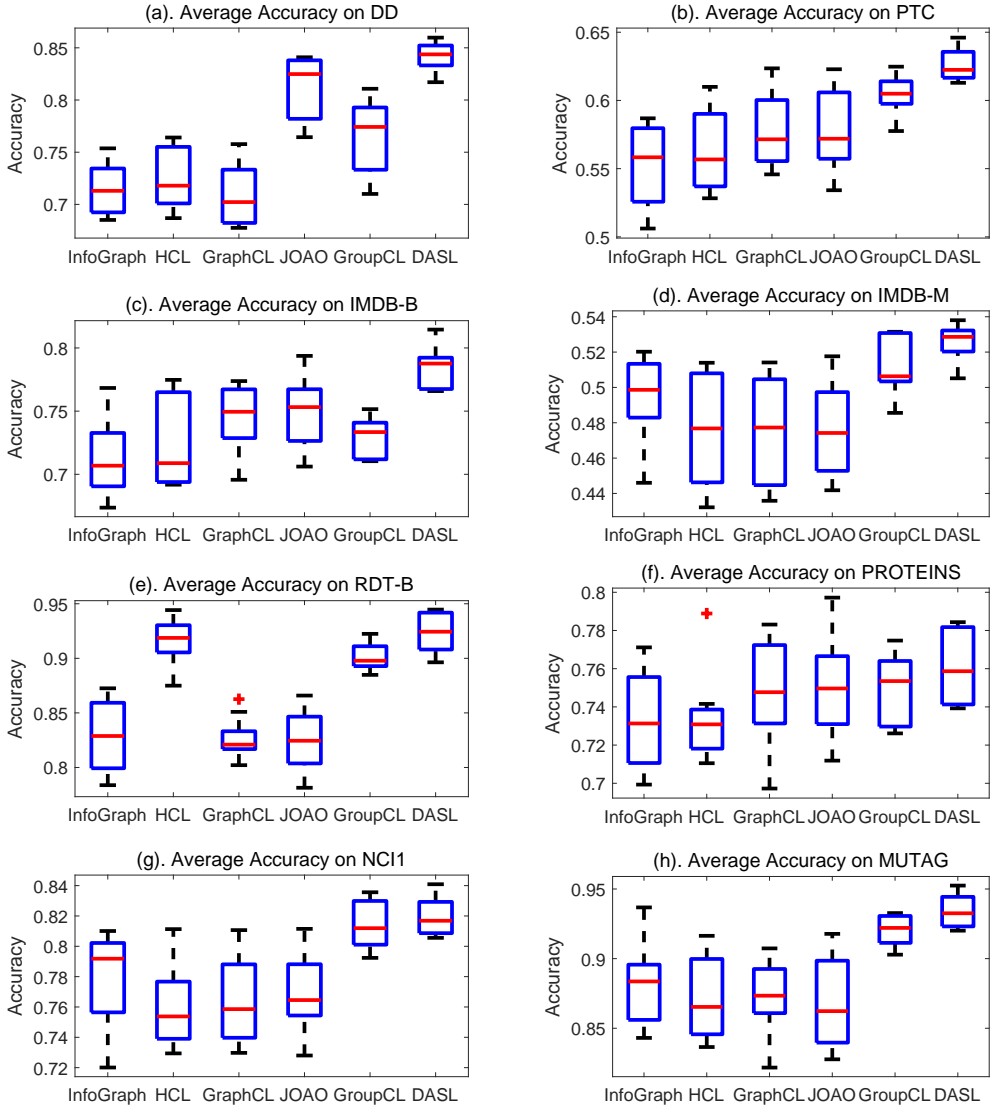

Figure 7: Accuracy rates of compared methods on traditional unsupervised graph embedding tasks including eight popular datasets.

approaches, our method does as well as GraphCL, JOAO, and GroupCL. Furthermore, for most of the eight datasets, our method can outperform the three methods with a higher accuracy mean and lower accuracy variance.

## A.3 VISUALIZATION OF THE TRAINING CURVES

In our experiments, we further record the values of empirical losses and regularization terms of the traditional methods (including the Npair loss with BN and the ProxyAnchor loss with ResNet-50) and our approach in each epoch.

We can find from Fig. 8 that the loss functions of traditional approaches can always decrease well and converge to stable values, but the inconsistencies among representation differences (i.e., the CTV norm values) are still large. After introducing the DAR term in our approach, the empirical loss can still converge to a stable value, and more importantly the corresponding CTV norm is effectively reduced and controlled, so that the final classification performance is successfully improved (as shown in Tab. 1).

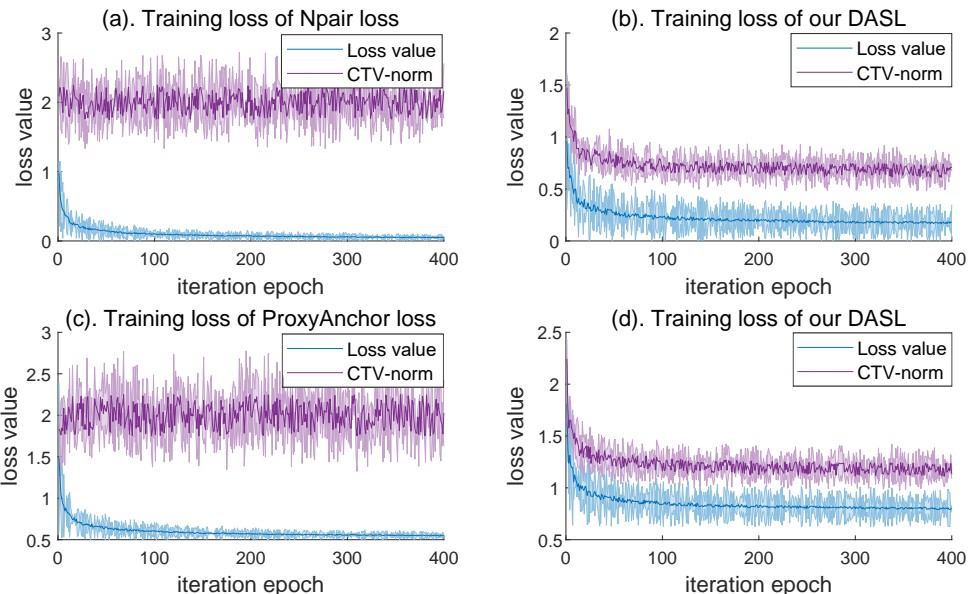

Figure 8: Training curves of the baseline methods and our method on the CAR dataset.

Table 8: Training time of the baseline methods and our proposed method (100 epochs, in hours).

| METHOD | CIFAR-10 | | ImageNet-100 | |
|---|---|---|---|---|
| | 512 | 1024 | 512 | 1024 |
| SimCLR (Chen et al., 2020) | 2.3 | 1.3 | 10.9 | 5.5 |
| SwAV (Caron et al., 2020) | 2.6 | 1.7 | 11.5 | 5.8 |
| DASL (SimCLR+DAR) | 2.5 | 1.6 | 11.3 | 5.9 |
| DASL (SwAV+DAR) | 2.6 | 1.9 | 11.8 | 5.9 |

## A.4 RUNNING TIME COMPARISON

In our regularization term, we adopted the $\ell_2$-norm to implement the measure function $\mathcal{G}(\cdot)$ of CTV norm. We would like to investigate if the efficiency of the learning algorithm will be affected by the calculation of CTV norm. Therefore, here we further provide experiments to record the training time of our method as well as the corresponding baseline method. Specifically, we use two NVIDIA TeslaV100 GPUs to train our method based on SimCLR and SwAV with 100 epochs, respectively. For each case, we set the batch size to $512$ and $1024$.

In Tab. 8, we can find that the proposed regularizer only brings in very little additional time consumption. This is because the gradient calculation of CTV norm is independent to the size of training data, so the training time is still acceptable in practice use.

## A.5 PARAMETRIC SENSITIVITY

Here we investigate the parametric sensitivity of the regularization parameter $\lambda$ in our method. Specifically, we change $\lambda$ in $[0.01, 5]$, and we record the classification accuracy of our method on STL-10 and CIFAR-10 datasets (batch size=256/512/1024, epochs=100). Tab. 9 shows that the accuracy variation of our method is smaller than $2\%$. These results clearly demonstrate that the regularization parameters $\lambda$ is very stable within a given range. It implies that the hyper-parameter of our method can be easily tuned in practice use.

Table 9: Parametric sensitivity of $\lambda$ on the STL-10 and CIFAR datasets (%). Here $\lambda$ is changed within $[0.01, 5]$.

| dataset (batchsize) | 0.01 | 0.1 | 0.5 | 1.5 | 5 |
|---|---|---|---|---|---|
| STL-10 (256) | 75.8 | 76.5 | **77.2** | 76.9 | 76.2 |
| STL-10 (512) | 77.6 | 78.3 | **79.5** | 79.5 | 78.2 |
| STL-10 (1024) | 79.9 | 80.5 | **81.3** | 80.9 | 80.5 |
| CIFAR-10 (256) | 87.5 | 87.5 | **88.3** | 88.1 | 87.5 |
| CIFAR-10 (512) | 89.7 | 90.7 | **91.2** | 91.0 | 90.8 |
| CIFAR-10 (1024) | 90.2 | 91.6 | 92.5 | **92.6** | 91.5 |

## A.6 CONVERGENCE ANALYSIS

We have the following theoretical result to investigate the convergence behavior of iteration points $\varphi^{(1)}, \varphi^{(2)}, \ldots, \varphi^{(T)}$ obtained by Eq. (7).

**Theorem 4.** *We assume that function $\mathcal{F}(\varphi)$ has a $\delta$-bounded gradient ($\|\nabla\mathcal{F}(\varphi)\|_2 < \delta$) and let $\eta = \sqrt{2(\mathcal{F}(\varphi^{(0)}) - \mathcal{F}(\varphi^*))/(S\delta^2 T)}$, and then for iteration points in Algorithm 1, we have $\min_{0 \leq t \leq T} \mathbb{E}[\|\nabla\mathcal{F}(\varphi^{(t)})\|_2] \leq \sqrt{2S\Delta\mathcal{F}/T}\delta$, where $\Delta\mathcal{F} = \mathcal{F}(\varphi^{(0)}) - \mathcal{F}(\varphi^*)$ and $S > 0$ is a Lipschitz constant such that $\|\nabla\mathcal{F}(\varphi) - \nabla\mathcal{F}(\varphi')\|_2 \leq S\|\varphi - \varphi'\|_2$.*

Notice that in the above theorem, the variables $S$, $\Delta\mathcal{F}$, and $\delta$ are all independent of $T$. The gradients of the iteration points of our final learning objective $\mathcal{F}(\varphi)$ will gradually decrease to 0. It means that the iteration points $\varphi^{(1)}, \varphi^{(2)}, \ldots, \varphi^{(T)}$ will converge to a stationary point of the learning objective $\mathcal{F}$ with a convergence rate $\mathcal{O}(1/\sqrt{T})$, where $T$ is the number of iterations.

# B PROOFS

## B.1 THE EQUIVALENCE BETWEEN EQ (5) AND EQ. (6)

According to the definition of CVT norm, we have that

$$
\begin{aligned}
&\mathcal{R}_{\text{align}}(\varphi; \mathscr{X}) \\
&= \mathbb{E}_{\{b_j\}_{j=0}^n}\left\{\left\|\left[\nabla_\varphi^{(1)}(\boldsymbol{x}_{b_0}, \boldsymbol{x}_{b_1}), \nabla_\varphi^{(1)}(\boldsymbol{x}_{b_1}, \boldsymbol{x}_{b_2}), \ldots, \nabla_\varphi^{(1)}(\boldsymbol{x}_{b_{n-1}}, \boldsymbol{x}_{b_n})\right]\right\|_{\text{ctv}}\right\} \\
&= \frac{1}{\mathrm{C}_N^{n+1}} \sum_{1 \leq i < j \leq N,\, 1 \leq k < l \leq N,\, (i,j) \neq (k,l),\, y_i \neq y_j,\, y_k \neq y_l} \mathrm{C}_N^{(n+1)-4} \mathcal{G}\left(\nabla_\varphi^{(1)}(\boldsymbol{x}_i, \boldsymbol{x}_j) - \nabla_\varphi^{(1)}(\boldsymbol{x}_k, \boldsymbol{x}_l)\right) \\
&= \frac{1}{\mathrm{C}_N^{n+1}} \sum_{1 \leq i < j \leq N,\, 1 \leq k < l \leq N,\, (i,j) \neq (k,l),\, y_i \neq y_j,\, y_k \neq y_l} \mathrm{C}_N^{(n+1)-4} \mathcal{G}\left(\nabla_\varphi^{(2)}(\boldsymbol{x}_i, \boldsymbol{x}_j, \boldsymbol{x}_k, \boldsymbol{x}_l)\right) \\
&= \frac{\mathrm{C}_N^{(n+1)-4}}{\mathrm{C}_N^{n+1}} \sum_{1 \leq i < j \leq N,\, 1 \leq k < l \leq N,\, (i,j) \neq (k,l),\, y_i \neq y_j,\, y_k \neq y_l} \mathcal{G}\left(\nabla_\varphi^{(2)}(\boldsymbol{x}_i, \boldsymbol{x}_j, \boldsymbol{x}_k, \boldsymbol{x}_l)\right) \\
&= 2\frac{\mathrm{C}_N^{(n+1)-4}}{\mathrm{C}_N^{n+1}} \left\|\left[\nabla_\varphi^{(1)}(\boldsymbol{x}_1, \boldsymbol{x}_2), \ldots, \nabla_\varphi^{(1)}(\boldsymbol{x}_i, \boldsymbol{x}_j), \ldots, \nabla_\varphi^{(1)}(\boldsymbol{x}_{N-1}, \boldsymbol{x}_N)\right]_{1 \leq i < j \leq N,\, y_i \neq y_j}\right\|_{\text{ctv}},
\end{aligned}
\tag{11}
$$

where the constant $\mathrm{C}_N^{(n+1)-4}/\mathrm{C}_N^{n+1}$ only depends on the batch size and sample size.

## B.2 DISCUSSION ON THE TRIVIAL SOLUTION $\mathcal{R}_{\text{ALIGN}}(\varphi; \mathscr{X}) = 0$

Here we would like to discuss if our learning algorithm will obtain the trivial solution $\mathcal{R}_{\text{align}}(\varphi; \mathscr{X}) = 0$. We prove that there exists $\lambda_0 > 0$ such that $\forall \lambda \in (0, \lambda_0)$, $\mathcal{R}_{\text{align}}(\varphi_\lambda^*; \mathscr{X}) > 0$,

where $\varphi_\lambda^* \in \arg\min\limits_{\varphi \in \mathcal{H}} \mathcal{L}_{\mathrm{emp}}(\varphi; \mathscr{X}) + \lambda \mathcal{R}_{\mathrm{align}}(\varphi; \mathscr{X})$. To be specific, we suppose that

$$\varphi_0^* \in \arg\min\limits_{\varphi \in \mathcal{H}} \mathcal{L}_{\mathrm{emp}}(\varphi; \mathscr{X}), \tag{12}$$

and by using the definition of empirical loss, we have that

$$\mathcal{R}_{\mathrm{align}}(\varphi_0^*; \mathscr{X}) = Q > 0. \tag{13}$$

We let $A(\lambda) = \mathcal{L}_{\mathrm{emp}}(\varphi; \mathscr{X}) + \lambda \mathcal{R}_{\mathrm{align}}(\varphi; \mathscr{X})$, and we have that $A(\lambda)$ is always continuous w.r.t. $\lambda$, and thus we have

$$\lim_{\lambda \to 0} A(\lambda) = A(0) = \mathcal{L}_{\mathrm{emp}}(\varphi_0^*; \mathscr{X}), \tag{14}$$

and

$$\lim_{\lambda \to 0} \mathcal{R}_{\mathrm{align}}(\varphi_\lambda^*; \mathscr{X}) = \mathcal{R}_{\mathrm{align}}(\varphi_0^*; \mathscr{X}) = Q. \tag{15}$$

By the property of a positive limitation, we have that there exists $\lambda_0 > 0$ such that $\forall \lambda \in (0, \lambda_0)$

$$\frac{Q}{2} \le \mathcal{R}_{\mathrm{align}}(\varphi_\lambda^*; \mathscr{X}) \le \frac{3Q}{2}, \tag{16}$$

where $\lambda_0$ is a sufficiently small number satisfying $|\mathcal{R}_{\mathrm{align}}(\varphi_\lambda^*; \mathscr{X}) - Q| \le \epsilon = \frac{Q}{2}$. It clearly reveals that the trivial solution will not be incurred by our learning algorithm with the controllable regularization parameter $\lambda$.

## B.3 PROOF FOR THEOREM 1

*Proof.* For a matrix $\boldsymbol{M} \in \mathbb{R}^{h \times H}$, the *non-negativity* of CTV norm can be directly obtained from the non-negative element $\mathcal{G}(\boldsymbol{M}_j - \boldsymbol{M}_k)$. For the *homogeneity*, we have that $\forall \mu \in \mathbb{R}$

$$
\begin{aligned}
&\|\mu \boldsymbol{M}\|_{\mathrm{ctv}} \\
&= \sum_{1 \le j < k \le H} \mathcal{G}(\mu \boldsymbol{M}_j - \mu \boldsymbol{M}_k) \\
&= \sum_{1 \le j < k \le H} \mathcal{G}(\mu(\boldsymbol{M}_j - \boldsymbol{M}_k)) \\
&= \sum_{1 \le j < k \le H} \mu \mathcal{G}((\boldsymbol{M}_j - \boldsymbol{M}_k)) \\
&= \mu \|\boldsymbol{M}\|_{\mathrm{ctv}},
\end{aligned}
\tag{17}
$$

where the third equation is based on the homogeneity of the measure function $\mathcal{G}(\cdot)$. Then, for the *triangle property*, we have that $\forall \boldsymbol{P}, \boldsymbol{Q} \in \mathbb{R}^{h \times H}$

$$
\begin{aligned}
&\|\boldsymbol{P} + \boldsymbol{Q}\|_{\mathrm{ctv}} \\
&= \sum_{1 \le j < k \le H} \mathcal{G}((\boldsymbol{P}_j + \boldsymbol{Q}_j) - (\boldsymbol{P}_k + \boldsymbol{Q}_k)) \\
&= \sum_{1 \le j < k \le H} \mathcal{G}((\boldsymbol{P}_j - \boldsymbol{P}_k) + (\boldsymbol{Q}_j - \boldsymbol{Q}_k)) \\
&= \sum_{1 \le j < k \le H} \mathcal{G}(\boldsymbol{P}_j - \boldsymbol{P}_k) + \mathcal{G}(\boldsymbol{Q}_j - \boldsymbol{Q}_k) \\
&= \sum_{1 \le j < k \le H} \mathcal{G}(\boldsymbol{P}_j - \boldsymbol{P}_k) + \sum_{1 \le j < k \le L} \mathcal{G}(\boldsymbol{Q}_j - \boldsymbol{Q}_k) \\
&= \|\boldsymbol{P}\|_{\mathrm{ctv}} + \|\boldsymbol{Q}\|_{\mathrm{ctv}},
\end{aligned}
\tag{18}
$$

which completes the proof. $\square$

## B.4 PROOF FOR THEOREM 2

**Lemma 1.** *For independent random variables $t_1, t_2, \ldots, t_n \in \mathcal{T}$ and a given function $\omega : \mathcal{T}^n \to \mathbb{R}$, if $\forall v_i' \in \mathcal{T}$ ($i = 1, 2, \ldots, n$), the function satisfies*

$$|\omega(t_1, \ldots, t_i, \ldots, t_n) - \omega(t_1, \ldots, t_i', \ldots, t_n)| \le \rho_i, \tag{19}$$

*then for any given $\mu > 0$, it holds that $P\{|\omega(t_1, \ldots, t_n) - \mathbb{E}[\omega(t_1, \ldots, t_n)]| > \mu\} \le 2\mathrm{e}^{-2\mu^2 / \sum_{i=1}^n \rho_i^2}$.*

*Proof.* Firstly, we denote that

$$\omega = \frac{1}{N} \sum_{\boldsymbol{t},\widehat{\boldsymbol{t}} \in \mathscr{X}} d_{\boldsymbol{\varphi}}(\boldsymbol{t},\widehat{\boldsymbol{t}}), \tag{20}$$

and

$$\widetilde{\omega_r} = \frac{1}{N} \left( \sum_{\boldsymbol{t},\widehat{\boldsymbol{t}} \in \mathscr{X}, (\boldsymbol{t},\widehat{\boldsymbol{t}}) \neq (\boldsymbol{t}_r,\widehat{\boldsymbol{t}}_r)} d_{\boldsymbol{\varphi}}(\boldsymbol{t},\widehat{\boldsymbol{t}}) + d_{\boldsymbol{\varphi}}(\boldsymbol{x},\widehat{\boldsymbol{x}}) \right), \tag{21}$$

where $\boldsymbol{x}$ and $\widehat{\boldsymbol{x}}$ obey the same distribution with the instances in $\mathscr{X}$. Then we have that

$$|\omega - \widetilde{\omega_r}|$$

$$= \frac{1}{N} \left| \sum_{\boldsymbol{t},\widehat{\boldsymbol{t}} \in \mathscr{X}, (\boldsymbol{t},\widehat{\boldsymbol{t}}) \neq (\boldsymbol{t}_r,\widehat{\boldsymbol{t}}_r)} d_{\boldsymbol{\varphi}}(\boldsymbol{t},\widehat{\boldsymbol{t}}) + d_{\boldsymbol{\varphi}}(\boldsymbol{x},\widehat{\boldsymbol{x}}) - \sum_{\boldsymbol{t},\widehat{\boldsymbol{t}} \in \mathscr{X}} d_{\boldsymbol{\varphi}}(\boldsymbol{t},\widehat{\boldsymbol{t}}) \right|$$

$$\leq \frac{1}{N} \left[ d_{\boldsymbol{\varphi}}(\boldsymbol{t}_r,\widehat{\boldsymbol{t}}_r) - d_{\boldsymbol{\varphi}}(\boldsymbol{x},\widehat{\boldsymbol{x}}) \right]$$

$$\leq \frac{L^{\frac{\mathcal{L}_{\text{emp}}(\boldsymbol{\varphi};\mathscr{X})}{\lambda}} \left( \|\boldsymbol{x} - \widehat{\boldsymbol{x}}\|_2 - \|\boldsymbol{t}_r,\widehat{\boldsymbol{t}}_r\|_2 \right) + \max\{d_{\boldsymbol{\varphi}}(\boldsymbol{t},\widehat{\boldsymbol{t}})|\boldsymbol{t},\widehat{\boldsymbol{t}} \in \mathscr{X}\})}{N}, \tag{22}$$

where $L > 0$ is the Lipschitz constant of $d_{\boldsymbol{\varphi}}$. Meanwhile, we have

$$\frac{1}{N} \sum_{\boldsymbol{t},\widehat{\boldsymbol{t}} \in \mathscr{X}} d_{\boldsymbol{\varphi}}(\boldsymbol{t},\widehat{\boldsymbol{t}}) - \mathbb{E} \left( \sum_{\boldsymbol{t},\widehat{\boldsymbol{t}} \in \mathscr{X}} d_{\boldsymbol{\varphi}}(\boldsymbol{t},\widehat{\boldsymbol{t}}) - \right)$$

$$= \overline{d}_{\boldsymbol{\varphi}}(\boldsymbol{t},\widehat{\boldsymbol{t}}) - d_{\boldsymbol{\varphi}}(\boldsymbol{x},\widehat{\boldsymbol{x}}). \tag{23}$$

By Lemma 1, we let that for all $i = 1, 2, \ldots, N$

$$\rho_i = \frac{L^{\frac{\mathcal{L}_{\text{emp}}(\boldsymbol{\varphi};\mathscr{X})}{\lambda}} \left( \|\boldsymbol{x} - \widehat{\boldsymbol{x}}\|_2 - \|\boldsymbol{t}_r,\widehat{\boldsymbol{t}}_r\|_2 \right) + \max\{d_{\boldsymbol{\varphi}}(\boldsymbol{t},\widehat{\boldsymbol{t}})|\boldsymbol{t},\widehat{\boldsymbol{t}} \in \mathscr{X}\})}{N}, \tag{24}$$

so that we have

$$P \left\{ \left| \mathcal{L}_{\text{emp}}(\boldsymbol{\varphi};\mathscr{X}) - \widetilde{\mathcal{L}}_{\text{emp}}(\boldsymbol{\varphi};\mathscr{D}) \right| < \frac{L^{\frac{\mathcal{L}_{\text{emp}}(\boldsymbol{\varphi};\mathscr{X})}{\lambda}} \left( \|\boldsymbol{x} - \widehat{\boldsymbol{x}}\|_2 - \|\boldsymbol{t}_r,\widehat{\boldsymbol{t}}_r\|_2 \right) + \max\{d_{\boldsymbol{\varphi}}(\boldsymbol{t},\widehat{\boldsymbol{t}})|\boldsymbol{t},\widehat{\boldsymbol{t}} \in \mathscr{X}\})}{N} \sqrt{\frac{\ln(2/\delta)}{2N}} \right\}$$

$$= 1 - 2e^{-2\mu^2/\sum_{i=1}^N \rho_i^2}$$

$$\geq 1 - 2e^{\frac{-2N(\eta\sqrt{[\ln(2/\delta)]/(2N)})^2}{\max^2((C+2/C)\omega(n)\log(1+\max\{d_{\boldsymbol{\varphi}}(\boldsymbol{t},\widehat{\boldsymbol{t}})|\boldsymbol{t} \in \mathscr{X}\}\alpha))}}$$

$$= 1 - 2e^{-2N\left(\sqrt{[\ln(2/\delta)]/(2N)}\right)^2}$$

$$= 1 - 2e^{-\ln(2/\delta)}$$

$$= 1 - \delta, \tag{25}$$

where $\eta = \frac{L^{\frac{\mathcal{L}_{\text{emp}}(\boldsymbol{\varphi};\mathscr{X})}{\lambda}} \left( \|\boldsymbol{x} - \widehat{\boldsymbol{x}}\|_2 - \|\boldsymbol{t}_r,\widehat{\boldsymbol{t}}_r\|_2 \right) + \max\{d_{\boldsymbol{\varphi}}(\boldsymbol{t},\widehat{\boldsymbol{t}})|\boldsymbol{t},\widehat{\boldsymbol{t}} \in \mathscr{X}\})}{N}$ and $\mu = \theta\sqrt{[\ln(2/\delta)]/(2N)}$. It means that have that with probability at least $1 - \delta$,

$$|d_{\boldsymbol{\varphi}}(\boldsymbol{x},\widehat{\boldsymbol{x}}) - \overline{d}_{\boldsymbol{\varphi}}(\boldsymbol{t},\widehat{\boldsymbol{t}})| \leq \frac{L^{\frac{\mathcal{L}_{\text{emp}}(\boldsymbol{\varphi};\mathscr{X})}{\lambda}} \left( \|\boldsymbol{x} - \widehat{\boldsymbol{x}}\|_2 - \|\boldsymbol{t}_r,\widehat{\boldsymbol{t}}_r\|_2 \right) + \max\{d_{\boldsymbol{\varphi}}(\boldsymbol{t},\widehat{\boldsymbol{t}})|\boldsymbol{t},\widehat{\boldsymbol{t}} \in \mathscr{X}\})}{N} \sqrt{\frac{\ln(2/\delta)}{2N}}. \tag{26}$$

Similarly, for another data pair $\boldsymbol{z}$ and $\widehat{\boldsymbol{z}}$ obey the same distribution with the instances in $\mathscr{X}$, we have that with probability at least $1 - \delta$,

$$|d_{\boldsymbol{\varphi}}(\boldsymbol{z},\widehat{\boldsymbol{z}}) - \overline{d}_{\boldsymbol{\varphi}}(\boldsymbol{t},\widehat{\boldsymbol{t}})| \leq \frac{L^{\frac{\mathcal{L}_{\text{emp}}(\boldsymbol{\varphi};\mathscr{X})}{\lambda}} \left( \|\boldsymbol{z} - \widehat{\boldsymbol{z}}\|_2 - \|\boldsymbol{t}_r,\widehat{\boldsymbol{t}}_r\|_2 \right) + \max\{d_{\boldsymbol{\varphi}}(\boldsymbol{t},\widehat{\boldsymbol{t}})|\boldsymbol{t},\widehat{\boldsymbol{t}} \in \mathscr{X}\})}{N} \sqrt{\frac{\ln(2/\delta)}{2N}}. \tag{27}$$

By combining the above two equations, we thus have

$$
\begin{aligned}
&|d_{\boldsymbol{\varphi}}(\boldsymbol{z}, \widehat{\boldsymbol{z}}) - d_{\boldsymbol{\varphi}}(\boldsymbol{x}, \widehat{\boldsymbol{x}})| \\
&= |d_{\boldsymbol{\varphi}}(\boldsymbol{z}, \widehat{\boldsymbol{z}}) - \overline{d}_{\boldsymbol{\varphi}}(\boldsymbol{t}, \widehat{\boldsymbol{t}}) + \overline{d}_{\boldsymbol{\varphi}}(\boldsymbol{t}, \widehat{\boldsymbol{t}}) - d_{\boldsymbol{\varphi}}(\boldsymbol{x}, \widehat{\boldsymbol{x}})| \\
&\leq |d_{\boldsymbol{\varphi}}(\boldsymbol{z}, \widehat{\boldsymbol{z}}) - \overline{d}_{\boldsymbol{\varphi}}(\boldsymbol{t}, \widehat{\boldsymbol{t}})| + |\overline{d}_{\boldsymbol{\varphi}}(\boldsymbol{t}, \widehat{\boldsymbol{t}}) - d_{\boldsymbol{\varphi}}(\boldsymbol{x}, \widehat{\boldsymbol{x}})| \\
&\leq \frac{L^{\frac{\mathcal{L}_{\text{emp}}(\boldsymbol{\varphi};\mathscr{X})}{\lambda}} \left( \|\boldsymbol{x} - \widehat{\boldsymbol{x}}\|_2 - \|\boldsymbol{t}_r, \widehat{\boldsymbol{t}}_r\|_2 + \|\boldsymbol{z} - \widehat{\boldsymbol{z}}\|_2 - \|\boldsymbol{t}_r, \widehat{\boldsymbol{t}}_r\|_2 \right) + \max\{d_{\boldsymbol{\varphi}}(\boldsymbol{t}, \widehat{\boldsymbol{t}})|\boldsymbol{t}, \widehat{\boldsymbol{t}} \in \mathscr{X}\}}{N} \sqrt{\frac{\ln(2/\delta)}{2N}} \\
&\leq \frac{L^{\frac{\mathcal{L}_{\text{emp}}(\boldsymbol{\varphi}^{(0)};\mathscr{X})}{\lambda}} \left( \|\boldsymbol{x} - \widehat{\boldsymbol{x}}\|_2 + \|\boldsymbol{z} - \widehat{\boldsymbol{z}}\|_2 \right) + \max\{d_{\boldsymbol{\varphi}}(\boldsymbol{t}, \widehat{\boldsymbol{t}})|\boldsymbol{t}, \widehat{\boldsymbol{t}} \in \mathscr{X}\}}{N} \sqrt{\frac{\ln(2/\delta)}{2N}} \\
&\leq \frac{\xi(\lambda) \left( \|\boldsymbol{x} - \widehat{\boldsymbol{x}}\|_2 + \|\boldsymbol{z} - \widehat{\boldsymbol{z}}\|_2 \right) + \max\{d_{\boldsymbol{\varphi}}(\boldsymbol{t}, \widehat{\boldsymbol{t}})|\boldsymbol{t}, \widehat{\boldsymbol{t}} \in \mathscr{X}\}}{N} \sqrt{\frac{\ln(2/\delta)}{2N}},
\end{aligned}
\tag{28}
$$

where $\xi(\lambda) = L^{\frac{\mathcal{L}_{\text{emp}}(\boldsymbol{\varphi}^{(0)};\mathscr{X})}{\lambda}}$ is monotonically decreasing w.r.t. $\lambda$. The proof is completed. $\qquad\square$

### B.5 PROOF FOR THEOREM 3

*Proof.* We prove Theorem 4 by analyzing the perturbation (i.e., $\rho_i$ in the above Eq. (19)) of the loss function $\mathcal{L}$.

We denote that

$$
\omega = \mathcal{L}_{\text{emp}}(\boldsymbol{\varphi}; \mathscr{X}) = \frac{1}{N} \sum_{i=1}^{N} -\log \frac{e^{-d_{\boldsymbol{\varphi}}(\boldsymbol{x}_i, \boldsymbol{x}^+)/\gamma}}{e^{-d_{\boldsymbol{\varphi}}(\boldsymbol{x}_i, \boldsymbol{x}^+)/\gamma} + \sum_{j=1}^{n} e^{-d_{\boldsymbol{\varphi}}(\boldsymbol{x}_i, \boldsymbol{x}_{b_j})/\gamma}},
\tag{29}
$$

and

$$
\widetilde{\omega_r} = \frac{1}{N} \left[ \left( \sum_{i \neq r}^{N} -\log \frac{e^{-d_{\boldsymbol{\varphi}}(\boldsymbol{x}_i, \boldsymbol{x}^+)/\gamma}}{e^{-d_{\boldsymbol{\varphi}}(\boldsymbol{x}_i, \boldsymbol{x}^+)/\gamma} + \sum_{j=1}^{n} e^{-d_{\boldsymbol{\varphi}}(\boldsymbol{x}_i, \boldsymbol{x}_{b_j})/\gamma}}, \right) - \log \frac{e^{-d_{\boldsymbol{\varphi}}(\widehat{\boldsymbol{x}}, \widehat{\boldsymbol{x}}^+)/\gamma}}{e^{-d_{\boldsymbol{\varphi}}(\widehat{\boldsymbol{x}}, \widehat{\boldsymbol{x}}^+)/\gamma} + \sum_{j=1}^{n} e^{-d_{\boldsymbol{\varphi}}(\widehat{\boldsymbol{x}}, \widehat{\boldsymbol{x}}_{b_j})/\gamma}} \right],
\tag{30}
$$

where $(\widehat{\boldsymbol{x}}, \{\widehat{\boldsymbol{x}}_{b_j}\}_{j=1}^{n})$ is an arbitrary mini-batch from the sample space. Then we have that

$$
\begin{aligned}
&|\omega - \widetilde{\omega_r}| \\
&= \frac{1}{N} \left| \log \frac{e^{-d_{\boldsymbol{\varphi}}(\widehat{\boldsymbol{x}}, \widehat{\boldsymbol{x}}^+)/\gamma}}{e^{-d_{\boldsymbol{\varphi}}(\widehat{\boldsymbol{x}}, \widehat{\boldsymbol{x}}^+)/\gamma} + \sum_{j=1}^{n} e^{-d_{\boldsymbol{\varphi}}(\widehat{\boldsymbol{x}}, \widehat{\boldsymbol{x}}_{b_j})/\gamma}} - \log \frac{e^{-d_{\boldsymbol{\varphi}}(\boldsymbol{x}_r, \boldsymbol{x}^+)/\gamma}}{e^{-d_{\boldsymbol{\varphi}}(\boldsymbol{x}_r, \boldsymbol{x}^+)/\gamma} + \sum_{j=1}^{n} e^{-d_{\boldsymbol{\varphi}}(\boldsymbol{x}_r, \boldsymbol{x}_{b_j})/\gamma}} \right| \\
&\leq \frac{1}{N} \log \left[ \frac{e^{-d_{\boldsymbol{\varphi}}(\widehat{\boldsymbol{x}}, \widehat{\boldsymbol{x}}^+)/\gamma} (e^{-d_{\boldsymbol{\varphi}}(\boldsymbol{x}_r, \boldsymbol{x}^+)/\gamma} + \sum_{j=1}^{n} e^{-d_{\boldsymbol{\varphi}}(\boldsymbol{x}_r, \boldsymbol{x}_{b_j})/\gamma})}{e^{-d_{\boldsymbol{\varphi}}(\boldsymbol{x}_r, \boldsymbol{x}^+)/\gamma} (e^{-d_{\boldsymbol{\varphi}}(\widehat{\boldsymbol{x}}, \widehat{\boldsymbol{x}}^+)/\gamma} + \sum_{j=1}^{n} e^{-d_{\boldsymbol{\varphi}}(\widehat{\boldsymbol{x}}, \widehat{\boldsymbol{x}}_{b_j})/\gamma})} \right] \\
&\leq \frac{(C + 2/C)\omega(n)\log(1 + \max\{d_{\boldsymbol{\varphi}}(\boldsymbol{t}, \widehat{\boldsymbol{t}})|\boldsymbol{t}, \widehat{\boldsymbol{t}} \in \mathscr{X}\})}{\alpha N},
\end{aligned}
\tag{31}
$$

where $\omega(n) = \log \left( \frac{e^2}{n} + 1 \right)$. Meanwhile, we have

$$
\begin{aligned}
&\frac{1}{N} \sum_{i=1}^{N} -\log \frac{e^{-d_{\boldsymbol{\varphi}}(\boldsymbol{x}_i, \boldsymbol{x}^+)/\gamma}}{e^{-d_{\boldsymbol{\varphi}}(\boldsymbol{x}_i, \boldsymbol{x}^+)/\gamma} + \sum_{j=1}^{n} e^{-d_{\boldsymbol{\varphi}}(\boldsymbol{x}_i, \boldsymbol{x}_{b_j})/\gamma}} - \mathbb{E} \left( -\log \frac{e^{-d_{\boldsymbol{\varphi}}(\boldsymbol{x}_i, \boldsymbol{x}^+)/\gamma}}{e^{-d_{\boldsymbol{\varphi}}(\boldsymbol{x}_i, \boldsymbol{x}^+)/\gamma} + \sum_{j=1}^{n} e^{-d_{\boldsymbol{\varphi}}(\boldsymbol{x}_i, \boldsymbol{x}_{b_j})/\gamma}} \right) \\
&= \mathcal{L}_{\text{emp}}(\boldsymbol{\varphi}; \mathscr{X}) - \widetilde{\mathcal{L}}_{\text{emp}}(\boldsymbol{\varphi}; \mathscr{D}).
\end{aligned}
\tag{32}
$$

By Lemma 1, we let that for all $i = 1, 2, \ldots, N$

$$
\rho_i = \frac{(C + 2/C)\omega(n)\log(1 + \max\{d_{\boldsymbol{\varphi}}(\boldsymbol{t}, \widehat{\boldsymbol{t}})|\boldsymbol{t}, \widehat{\boldsymbol{t}} \in \mathscr{X}\})}{\alpha N},
\tag{33}
$$

so that we have

$$P\left\{\left|\mathcal{L}_{\text{emp}}(\boldsymbol{\varphi};\mathscr{X}) - \widetilde{\mathcal{L}}_{\text{emp}}(\boldsymbol{\varphi};\mathscr{D})\right| < \frac{(C+2/C)\omega(n)\log(1+\max\{d_{\boldsymbol{\varphi}}(\boldsymbol{t},\widehat{\boldsymbol{t}})|\boldsymbol{t},\widehat{\boldsymbol{t}}\in\mathscr{X}\})}{\alpha}\sqrt{\frac{\ln(2/\delta)}{2N}}\right\}$$

$$= 1 - 2\mathrm{e}^{-2\mu^2/\sum_{i=1}^{N}\rho_i^2}$$

$$\geq 1 - 2\mathrm{e}^{\frac{-2N(\eta\sqrt{[\ln(2/\delta)]/(2N)})^2}{\max^2((C+2/C)\omega(n)\log(1+\max\{d_{\boldsymbol{\varphi}}(\boldsymbol{t},\widehat{\boldsymbol{t}})|\boldsymbol{t}\in\mathscr{X}\})\alpha)}}$$

$$= 1 - 2\mathrm{e}^{-2N\left(\sqrt{[\ln(2/\delta)]/(2N)}\right)^2}$$

$$= 1 - 2\mathrm{e}^{-\ln(2/\delta)}$$

$$= 1 - \delta, \tag{34}$$

where $\eta = \frac{(C+2/C)\omega(n)\log(1+\max\{d_{\boldsymbol{\varphi}}(\boldsymbol{t},\widehat{\boldsymbol{t}})|\boldsymbol{t}\in\mathscr{X}\})}{\alpha}$ and $\mu = \theta\sqrt{[\ln(2/\delta)]/(2N)}$. The proof is completed. $\square$

### B.6 PROOF FOR THEOREM 4

*Proof.* Firstly, by using the Lipschitz continuity of $\mathcal{F}(\boldsymbol{\varphi})$ we have that

$$\mathbb{E}[\mathcal{F}(\boldsymbol{\varphi}^{(t+1)})] - \mathbb{E}[\mathcal{F}(\boldsymbol{\varphi}^{(t+1)})]$$
$$\leq \mathbb{E}[\|\nabla\mathcal{F}(\boldsymbol{\varphi}^{(t+1)}) - (\boldsymbol{\varphi}^{(t+1)} - \boldsymbol{\varphi}^{(t)})\|_2^2 + S/2\|\boldsymbol{\varphi}^{(t+1)} - \boldsymbol{\varphi}^{(t)}\|_2^2]$$
$$\leq -\eta_t\mathbb{E}[\|\nabla\mathcal{F}(\boldsymbol{\varphi}^{(t)})\|_2^2] + (S\eta_t^2/2)\mathbb{E}[\|\nabla\mathcal{F}_{b_i}(\boldsymbol{\varphi}^{(t)})\|_2^2]$$
$$\leq -\eta_t\mathbb{E}[\|\nabla\mathcal{F}(\boldsymbol{\varphi}^{(t)})\|_2^2] + (S\eta_t^2/2)\delta^2, \tag{35}$$

where the second inequality follows from the fact that $\boldsymbol{\varphi}^{(t+1)}$ is updated by Algorithm 1. Then, we have that

$$\mathbb{E}[\|\nabla\mathcal{F}(\boldsymbol{\varphi}^{(t+1)})\|_2^2] \leq (1/\eta_t)\mathbb{E}[\mathcal{F}(\boldsymbol{\varphi}^{(t)}) - \mathcal{F}(\boldsymbol{\varphi}^{(t+1)})] + (L\eta_t/2)\delta^2, \tag{36}$$

and thus

$$\begin{cases} \mathbb{E}[\|\nabla\mathcal{F}(\boldsymbol{\varphi}^{(0)})\|_2^2] \leq (1/\eta_0)\mathbb{E}[\mathcal{F}(\boldsymbol{\varphi}^{(0)}) - \mathcal{F}(\boldsymbol{\varphi}^{(1)})] + (S\eta_0/2)\delta^2, \\ \mathbb{E}[\|\nabla\mathcal{F}(\boldsymbol{\varphi}^{(1)})\|_2^2] \leq (1/\eta_1)\mathbb{E}[\mathcal{F}(\boldsymbol{\varphi}^{(1)}) - \mathcal{F}(\boldsymbol{\varphi}^{(2)})] + (S\eta_1/2)\delta^2, \\ ... \\ \mathbb{E}[\|\nabla\mathcal{F}(\boldsymbol{\varphi}^{(T-1)})\|_2^2] \leq (1/\eta_{T-1})\mathbb{E}[\mathcal{F}(\boldsymbol{\varphi}^{(T-1)}) - \mathcal{F}(\boldsymbol{\varphi}^{(T)})] + (S\eta_{T-1}/2)\delta^2. \end{cases} \tag{37}$$

Finally, we combine all inequalities in the above Eq. (37) by letting $\eta_0 = \eta_1 = \cdots = \eta_{T-1} = \eta$. Then we have

$$\min_{0\leq t\leq T-1}\mathbb{E}[\|\nabla\mathcal{F}(\boldsymbol{\varphi}^{(t)})\|_2]$$
$$\leq \frac{1}{T}\sum_{t=0}^{T-1}\mathbb{E}[\|\nabla\mathcal{F}(\boldsymbol{\varphi}^{(t)})\|] + (S\eta/2)\delta^2$$
$$\leq \frac{1}{T\eta}\mathbb{E}[\mathcal{F}(\boldsymbol{\varphi}^{(0)}) - \mathcal{F}(\boldsymbol{\varphi}^{(t)})] + (S\eta/2)\delta^2$$
$$\leq \frac{1}{T\eta}(\mathcal{F}(\boldsymbol{\varphi}^{(0)}) - \mathcal{F}(\boldsymbol{\varphi}^*)) + (S\eta/2)\delta^2$$
$$\leq \frac{1}{\sqrt{T}}((\mathcal{F}(\boldsymbol{\varphi}^{(0)}) - \mathcal{F}(\boldsymbol{\mathcal{Q}}^*,\boldsymbol{\varphi}^*))/c + (Sc/2)\delta^2), \tag{38}$$

where $c = \eta\sqrt{T}$. We set $c = \sqrt{2(\mathcal{F}(\boldsymbol{\varphi}^{(0)}) - \mathcal{F}(\boldsymbol{\varphi}^*))/(S\delta^2)}$, and we have

$$\min_{0\leq t\leq T-1}\mathbb{E}[\|\nabla\mathcal{F}(\boldsymbol{\varphi}^{(t)})\|_2] \leq \sqrt{2S(\mathcal{F}(\boldsymbol{\varphi}^{(0)}) - \mathcal{F}(\boldsymbol{\varphi}^*))/T}\delta, \tag{39}$$

which completes the proof. $\square$

