# OpenReview forum: "Robust Similarity Learning with Difference Alignment Regularization"
_ICLR.cc/2024/Conference — ICLR 2024 poster_

### Official Review · Reviewer_V3dP · 2023-10-22

**Soundness:** 4 excellent
**Presentation:** 4 excellent
**Contribution:** 4 excellent
**Rating:** 8
**Confidence:** 5

**Summary:**

This paper addresses similarity-based representation learning, which is important to the machine learning field. In this paper, the authors propose a novel difference alignment regularization (DAR) to encourage all representation differences between inter-class instances to be as close as possible so that the learning algorithm can produce consistent differences to distinguish data points. The proposed method is sound with theoretical proof. The proposed method is extensively verified on various representation learning tasks, including supervised metric learning and unsupervised contrastive learning.

**Strengths:**

1. This paper addresses similarity-based representation learning, which is of great importance to the machine learning field.
2. The proposed method is sound with theoretical proof.
3. The proposed method is extensively verified on various representation learning tasks, including supervised metric learning and unsupervised contrastive learning.

**Weaknesses:**

1. Calculating the higher-order difference requires complex calculation complexity. It would be better to report the running time comparison of different methods with some specific dataset to show the effectiveness.
2. There are also other ways to enhance inter-class compactness in the metric learning field, e.g., [1-3]. Please discuss the differences between the current algorithm and these works.
3. It would be better to report the sensitivity with regard to the regularization parameter lambda. Does the change of it heavily influence the final performance? What is the default choice?

**Questions:**

Please refer to the weakness part.

Overall, this paper tackles the essential problem with a sound method. The proposed method is theoretically proven to have tight bounds, which is also evaluated with extensive experimental results in different metric learning settings. I am positive about this submission; my initial rating is “8”.

[1] Learning multiple local metrics: Global consideration helps. TPAMI 2019

[2] Deep metric learning with angular loss. ICCV 2017

[3] What Makes Objects Similar: A Unified Multi-Metric Learning Approach. TAPMI 2018

---

> ### Author Response · Authors · 2023-11-17
> **Response to Reviewer V3dP**
>
> Thank you for your positive and constructive comments! Our point-by-point responses are given below.
>
> ---
>
> **Comment_1:** Calculating the higher-order difference requires complex calculation complexity. It would be better to report the running time comparison of different methods with some specific dataset to show the effectiveness.
>
> **Response_1:** Thank you for the great suggestion! Here we would like to provide the training time comparison of our DASL and the baseline methods (SimCLR and SwAV) on the CIFAR-10 and ImageNet-100 datasets.  Specifically, we use two NVIDIA TeslaV100 GPUs to train our models based on SimCLR and SwAV with 100 epochs, respectively. For each case, we set the batch size to 512 and 1024.
>
> | Method | &nbsp; CIFAR-10 (512) &nbsp; | &nbsp; CIFAR-10 (1024) &nbsp; | &nbsp; ImageNet-100 (512) &nbsp; | &nbsp; ImageNet-100 (1024) &nbsp; |
> | :--- | :----: | :----: | :----: | :----: |
> | SimCLR | 2.3 | 1.3 | 10.9 | 5.5 |
> | SwAV | 2.6 | 1.7 | 11.5 | 5.8 |
> | DASL (SimCLR + DAR) | 2.6 | 1.6 | 11.3 | 5.9 |
> | DASL (SwAV + DAR) | 2.6 | 1.9 | 11.8 | 5.9 |
>
> The table above reveals that the introduction of the proposed regularizer causes very little increase in time consumption. This is because the gradient calculation of the CTV norm is independent of the size of the training data, thereby keeping the training time well within practical limits. We have also added the above time comparison in the revised appendix.
>
> ---
>
> **Comment_2:** There are also other ways to enhance inter-class compactness in the metric learning field, e.g., [1-3]. Please discuss the differences between the current algorithm and these works. [1] Learning multiple local metrics: Global consideration helps, TPAMI 2019. [2] Deep metric learning with angular loss, ICCV 2017. [3] What Makes Objects Similar: A Unified Multi-Metric Learning Approach, TAPMI 2018.
>
> **Response_2:** Thank you for the comment! We agree with the reviewer that the three works [1-3] also consider the inter-class similarities. However, our work differs substantially from the above three works, as here [1] and [3] mainly focus on learning a *global metric* merged from multiple *local metrics* to better capture the inter-class similarities in different contextual scenarios. Furthermore, [2] constructs a new *loss function* to mitigate the *incorrect* reduction of inter-class distances caused by the scale variance, and it does not consider the variation of the *correct* inter-class distances. In comparison, we propose a *general regularization technique* to encourage those correct inter-class differences to be as close as possible. These distinctions set our method apart from the three mentioned methods in terms of both motivation and methodology.
>
> We have cited the above three works and also added the corresponding discussions in our revised paper.
>
> ---
>
> **Comment_3:** It would be better to report the sensitivity with regard to the regularization parameter lambda. Does the change of it heavily influence the final performance? What is the default choice?
>
> **Response_3:** Thank you for the suggestion! In our appendix, we have already provided the corresponding description and discussion for the default value and sensitivity of the regularization parameter $\lambda$.
>
> To be more specific, we recommend a value of $\lambda$ equal to 0.5, as it consistently provides the best performance across a wide range of tasks. Additionally, we conducted experiments on STL-10 and CIFAR-10 datasets to thoroughly investigate the parametric sensitivity of $\lambda$. We systematically varied $\lambda$ within the range of [0.01, 5] and recorded the classification accuracy of our method under different conditions (batch sizes of 256, 512, and 1024, and 100 training epochs). The table presenting the results shows that the accuracy of our method exhibits a variation of less than 2% across these different parameter settings.
>
> | Dataset (batchsize) | &nbsp; $\lambda=0.01$ &nbsp; | &nbsp; $\lambda=0.1$ &nbsp; | &nbsp; $\lambda=0.5$ &nbsp; | &nbsp; $\lambda=1.5$ &nbsp; | &nbsp; $\lambda=5$ &nbsp; |
> | :--- | :----: | :----: | :----: | :----: | :----: |
> | STL (batchsize=256) | 75.8 | 76.5 | **77.2** | 76.9 | 76.2 |
> | STL (batchsize=512) | 77.6 | 78.3 | **79.5** | **79.5** | 78.2 |
> | STL (batchsize=1024) | 79.9 | 80.5 | **81.3** | 80.9 | 80.5 |
> | CIFAR-10 (batchsize=256) |  87.5 | 87.5 | **88.3** | 88.1 | 87.5 |
> | CIFAR-10 (batchsize=512) | 89.7 | 90.7 | **91.2** | 91.0 | 90.8 |
> | CIFAR-10 (batchsize=1024) | 90.2 | 91.6 | 92.5 | **92.6** | 91.5 |
>
> These results clearly demonstrate that the regularization parameter $\lambda$ is very stable within a given range, and thus the hyper-parameter of our method can be easily tuned in practical use.

---

> > ### Comment · Reviewer_V3dP · 2023-11-19
> > **Response to the authors**
> >
> > I thank the authors for providing a detailed rebuttal. For now, all my concerns are properly addressed, and I tend to maintain my initial rating of "8."

---

> > > ### Author Response · Authors · 2023-11-21
> > > **Thanks**
> > >
> > > Dear Reviewer V3dP,
> > >
> > > Thanks a lot for your positive comments!
> > >
> > > Best wishes,
> > >
> > > Authors of Paper ID 2934

---

### Official Review · Reviewer_YcLH · 2023-10-25

**Soundness:** 3 good
**Presentation:** 3 good
**Contribution:** 3 good
**Rating:** 8
**Confidence:** 4

**Summary:**

The authors propose to regularize ERM for similarity learning by enforcing consistency among distances between negative pairs of samples. Forcing the distances between negative pairs to be consistent leads to a more robust representation, and the authors demonstrate this theoretically using a PAC learning style upper bound on the generalization gap.

**Strengths:**

- Results are quite extensive and impressive.
- I found the method interesting. I like the way it was presented. However, I do have some reservations, as stated in the weaknesses and questions.
- Figures are very easy to understand.

**Weaknesses:**

- Regarding the method, could you give some insight as to why it's different from the standard contrastive loss? The standard contrastive loss is:

$L = ReLu(d^+ - \delta^+) + ReLu(\delta^- - d^-)$

where $d^+$ denotes the distance between positive pairs, $d^-$ denotes the distance between negative pairs, $\delta^+$ denotes the positive margin and $\delta^-$ denotes the native margin. This loss repels features of negative pairs until they are $\delta^-$ away from each-other. In effect, the contrastive loss ensures that all negative pairs are $\delta^-$ away from each-other. This would also satisfy your loss function. Could your loss function be doing something similar to enforcing a negative margin?

- There are some minor issues I saw with the math and the theorems that I don't think are consequential. For example, the notation is quite complicated; I think Algorithm 1 is obvious given the optimization objective in Eq. (8); Theorem 2 seems to trivially follow from the standard GD convergence result; Theorem 4 ignores the tradeoff between generalization and approximation (you say that higher lambda leads to a smaller upper-bound on the generalization gap, but a higher lambda also means $\mathcal{L}(\phi)$ is a bad approximation of what you actually want to minimize.) Obviously, it's not hard to design a loss function that generalizes well if you don't care how well it approximates the actual minimization problem; so I'm not sure how meaningful Theorem 4 is.

If there were a seven, I would give the paper a 7.

**Questions:**

See above.

---

> ### Author Response · Authors · 2023-11-17
> **Response to Reviewer YcLH**
>
> Thank you for your positive and constructive comments! Our point-by-point responses are provided below.
>
> ---
>
> **Comment_1:** Could you give some insight as to why it's different from the standard contrastive loss? The standard contrastive loss is $L=ReLu(d^{+}-\delta^{+})+ReLu(\delta^{-}-d^{-})$, where $d^{+}$ denotes the distance between positive pairs, $d^{-}$ denotes the distance between negative pairs, $\delta^{+}$ denotes the positive margin and $\delta^{-}$ denotes the native margin. This loss repels features of negative pairs until they are $\delta^{-}$ away from each other. This would also satisfy your loss function. Could your loss function be doing something similar to enforcing a negative margin?
>
> **Response_1:** As the reviewer suggested, the conventional contrastive similarity loss can indeed ensure that all negative pairs are $\delta^{-}$-away from each other, i.e., the inter-class distances are always larger than $\delta^{-}$. However, this effect is still quite different from our primary goal. Although the ReLu function can effectively penalize the incorrect inter-class distances $d^{-}$ that are smaller than $\delta^{-}$, it is still unclear how the correct inter-class distances are distributed in $(\delta^{-}, +\infty)$ because there is no constraint any more for these correct inter-class distances which are larger than $\delta^{-}$ (i.e., $ReLu(\delta^{-}-d^{-})$ is always 0). Therefore, we still need *difference alignment regularization* (DAR) to explicitly promote consistent representation differences.
>
> Furthermore, in our original manuscript, we provided experiments to demonstrate that our DAR can significantly improve the representative contrastive loss ProxyAnchor on the CAR, CUB, and SOP datasets. Meanwhile, the numerical experiments in Fig. 8 of Appendix A.3 also validate that the conventional ProxyAnchor fails to obtain the difference-aligned features. During the training phase, the CTV-norm of ProxyAnchor is generally located around a relatively large value. In comparison, our method can minimize the empirical loss under the condition of a very small CTV-norm value.
>
> In summary, our ultimate learning objective is different from directly enforcing a negative margin by contrastive loss, because we can further constrain those correct inter-class distances that lie in $(\delta^{-}, +\infty)$.
>
> ---
>
> **Comment_2:** Minor issues: complicated notation and trivial description of Theorem 2.
>
> **Response_2:** Thank you for pointing out the above problems! We have tried our best to further simplify the notation, and we will also move Theorem 2 to Appendix in the camera-ready version.
>
> ---
>
> **Comment_3:** Minor issue: Theorem 4 ignores the tradeoff between generalization and approximation (you say that higher lambda leads to a smaller upper-bound on the generalization gap, but a higher lambda also means $\mathcal{L}$ is a bad approximation of what you actually want to minimize.)
>
> **Response_3:** Yes, we agree with the reviewer that the tradeoff between *model fitting* and *model generalization* is a longstanding and critically important problem in machine learning. In our case, it is also very difficult to provide theoretical analyses to answer the optimal value of $\lambda$. However, we had provided empirical investigations to show that the modestly large $\lambda$ does not harm the fitting result and also obtains a good empirical risk $\mathcal{L}$.
>
> Specifically, from Fig. 8 we can clearly observe that the empirical loss $\mathcal{L}$ (for model fitting) can still be well minimized and converge to a stable point even though the regularization parameter $\lambda$ is increased from 0 to 0.5. It means that the value of $\mathcal{L}$ in Theorem 4 can still be kept at a small value when we increase the parameter $\lambda$. In this case, the shrinking of the error bound will give a good generalization ability (i.e., the small $\widetilde{\mathcal{L}}$).

---

> > ### Comment · Reviewer_YcLH · 2023-11-20
> > **Response to response**
> >
> > I thank the authors for their response.
> >
> > Looking to my original review, my only major concern was some sort of comparison to the standard contrastive loss. I find the authors' response to this concern to be sufficient.
> >
> > The paper is good and I raised my score to 8.
> >
> > As an aside, I personally find this style of writing to be unnecessarily complicated (between the complicated notation and theorems). I don't think the theorems (except maybe the last one) add much value to the paper. The idea is quite simple and effective once I understood the method, and I'm afraid the complicated way in which the idea is presented might scare away people who might use this.

---

> > > ### Author Response · Authors · 2023-11-21
> > > **Thank you for your positive comments**
> > >
> > > Dear Reviewer YcLH,
> > >
> > > Thank you very much for raising the score!
> > >
> > > We'd like to express our sincere gratitude for your thoughtful and thorough review of our paper.
> > >
> > > Best wishes,
> > >
> > > Authors of Paper ID 2934

---

### Official Review · Reviewer_Q8tk · 2023-11-01

**Soundness:** 2 fair
**Presentation:** 2 fair
**Contribution:** 2 fair
**Rating:** 5
**Confidence:** 5

**Summary:**

The paper presents a new approach to representation learning by emphasizing higher-order feature distances. It introduces Difference Alignment Regularization (DAR) to maintain consistent representation differences between inter-class instances, addressing the overfitting issue common in traditional methods. A novel cross-total-variation (CTV) norm is proposed to quantify these differences, and is simplified for optimization. DAR is integrated into Difference-Aligned Similarity Learning (DASL), reducing overfitting and enhancing performance. Theoretical and empirical analyses confirm DAR's effectiveness in tightening error bounds and improving performance in various learning tasks.

**Strengths:**

1.     Approach: The paper learns robust and generalized representation by focusing on reducing the the higher-order feature distances rather than the first-order differences, which is hardly explored before.
2.	Theoretical Justification: The paper not only proposes a new method but also provides theoretical evidence to support its effectiveness.
3.	Empirical Validation: The superiority of the Difference-Aligned Similarity Learning (DASL) method is backed by experiments conducted on multi-domain data (Image retrieval, face recognition, image classification and NLP classification), showing its effectiveness in both supervised metric learning and unsupervised contrastive learning tasks.

**Weaknesses:**

1.	Inconsistency between theoretical justification and method. Theorem 4 presents a bound on the generalization gap based on max{d'(t; t_hat)}, a metric akin to first-order measurement of feature distance. Despite the notion that smoothness of discrepancy could minimize the largest max{d'(t; t_hat)} by pulling it towards the majority, this mechanism inherently relies on first-order measurement. Consequently, the theorem does not robustly support the paper's primary claim, casting doubts on the reliability of the proposed solution.

2.	Inconsistency between motivation and method. Figure 2 suggests that a smoother distance between different distributions correlates with better performance. However, this conclusion appears flawed. The generalization in a mixed Gaussian scenario is predominantly governed by the distance of the distributions (a first-order attribute) rather than higher-order terms. For instance, three Gaussians with unit variance located at (0,1), (1, 0), and (1,10000) would exhibit a smaller generalization error compared to them being positioned at (0,1), (1, 0), and (1,1). The implication is that sufficient distance, regardless of higher-order distances, should invariably simplify distinguishability.

3.	Writing issues. I would suggestion the author revise the introduction, mainly on discuss the motivation and purpose of using higher order feature distance, rather than general discussion on representation learning.

**Questions:**

Please see above.

---

> ### Author Response · Authors · 2023-11-17
> **Response to Reviewer Q8tk (part 1/2)**
>
> Thank you for your appreciation of the novelty, theoretical analyses, and experimental results of our paper! Thanks also for your very insightful and constructive suggestions! Our point-by-point responses are as follows.
>
> ---
>
> **Comment_1:** Inconsistency between theoretical justification and method. Theorem 4 presents a bound on the generalization gap based on $\max{d_{\boldsymbol{\varphi}}(\boldsymbol{t}, \widehat{\boldsymbol{t}})}$, a metric akin to first-order measurement of feature distance. Despite the notion that smoothness of discrepancy could minimize the largest $d_{\boldsymbol{\varphi}}(\boldsymbol{t}, \widehat{\boldsymbol{t}})$ by pulling it towards the majority, this mechanism inherently relies on first-order measurement.  Consequently, the theorem does not robustly support the paper's primary claim.
>
> **Response_1:** First of all, we agree with the reviewer that the error bound in Theorem 4 is partly based on the first-order measurement (i.e., the distance $d_{\boldsymbol{\varphi}}(\boldsymbol{t}, \widehat{\boldsymbol{t}})$), and the higher-order difference is also defined as the consistency between first-order differences, so minimizing our CTV-norm regularizer will naturally affect the first-order difference. However, this does not mean that the existing first-order approaches can directly achieve the difference-aligned effect, because they do not explicitly constrain the variation of pairwise differences. In this case, as an important factor of the error bound, $\max{d_{\boldsymbol{\varphi}}(\boldsymbol{t}, \widehat{\boldsymbol{t}})}$ tends to become a large value which increases the corresponding generalization error.
>
> In comparison, by introducing our CTV-norm regularizer, the feature embedding is learned from a shrunken hypothesis space (i.e., the regularized learning objective $\mathcal{F}(\boldsymbol{\varphi})$ as we mentioned in Theorem 4), so that $\max{d_{\boldsymbol{\varphi}}(\boldsymbol{t}, \widehat{\boldsymbol{t}})}$ is well reduced to produce a tighter error bound. Meanwhile, we want to further clarify that the error bound in Theorem 4 not only depends on the maximal distance $\max{d_{\boldsymbol{\varphi}}(\boldsymbol{t}, \widehat{\boldsymbol{t}})}$, but is also influenced by the parameter $\lambda$. The introduction of our new regularizer will increase $\lambda$ from 0 to a default value of 0.5, and thus further reducing the final error bound.
>
> We agree with the reviewer that there is usually a real gap between the theoretical results and the effectiveness of the method, and such a gap commonly exists in many existing theoretical analyses of machine learning. In our case, we simply follow the common practice of investigating the upper bound of the generalization error, so that we can have a guarantee for the worst case of model generalizability.

---

> > ### Author Response · Authors · 2023-11-17
> > **Response to Reviewer Q8tk (part 2/2)**
> >
> > **Comment_2:** Inconsistency between motivation and method. Fig. 2 suggests that a smoother distance between different distributions correlates with better performance. However, this conclusion appears flawed. The generalization in a mixed Gaussian scenario is predominantly governed by the distance of the distributions (a first-order attribute) rather than higher-order terms. For instance, three Gaussians with unit variance located at (0,1), (1, 0), and (1,10000) would exhibit a smaller generalization error compared to them being positioned at (0,1), (1,0), and (1,1). The implication is that sufficient distance, regardless of higher-order distances, should invariably simplify distinguishability.
> >
> > **Response_2:** Thank you for providing the intuitive example! However, here we would like to clarify that in Fig. 2(a) and (b), our method will not change the cluster means from {(0,1), (1,0), (1,10000)} to {(0,1), (1,0), (1,1)} (pulling in cluster-3 while leaving the other two clusters fixed). This is because our representation differences are aligned under the condition of a small empirical risk, i.e., our ultimate learning objective *jointly* optimizes the loss function for discriminating data as well as the regularizer for difference alignment. It implies that when we pull in cluster-3, we have to further push away cluster-1 and cluster-2 (instead of keeping the two clusters fixed) to satisfy the constraint of our learning objective. Therefore, their exact distribution means are actually from {(0,9), (9,0), (35,35)} to {(0,12), (12,0), (20,20)}, as we discussed in Appendix A.1.
> >
> > The toy example in Fig. 2 fits well with both our motivation and methodology. Please notice that our fundamental motivation is to learn the difference-aligned feature embedding under the condition that the training data is well distinguished. To achieve this, our learning objective is not solely a single regularization term. Instead, we still have a basic empirical risk (which plays the role of *data discrimination*) to constrain the learned feature to fit the corresponding (pseudo) supervision. Our numerical experiments (in Fig. 8 of Appendix A.3) also reveal that the regularization term can be successfully minimized while ensuring that the empirical loss is a very small value. This suggests that there is still room for improving the representations with good distinguishability learned from the existing empirical loss (e.g., ProxyAnchor loss, Npair loss, and NCE loss in our experiments).
> >
> > ---
> >
> > **Comment_3:** I would suggest the author revise the introduction, mainly on discuss the motivation and purpose of using higher order feature distance, rather than general discussion on representation learning.
> >
> > **Response_3:** Thanks for your great suggestion! We have followed the reviewer’s suggestion to further polish our paper by discussing more the purpose of using higher-order differences. We hope you are satisfied with the updated introduction.
> >
> > ---
> >
> > Thanks a lot again for your insightful and constructive comments! Hope our clarification could address your concerns. We would be grateful if the reviewer could provide more suggestions to further improve the clarity of our paper.

---

### Official Review · Reviewer_SsFX · 2023-11-01

**Soundness:** 4 excellent
**Presentation:** 3 good
**Contribution:** 4 excellent
**Rating:** 6
**Confidence:** 5

**Summary:**

This work proposed a new unified regularization technique for both supervised and unsupervised similarity learning. The proposed regularization encourages the feature difference between inter-class samples to be similar. It is claimed to help avoid learning ill-posed clustering in the feature space, which is the often case in normal metric learning.

The authors did extensive numerical experiments on many tasks and datasets. The authors also provided experiments to show the sensitivity of the proposed method to hyperparameters. Results showed that the proposed regularization is easy to tune. Numerical results on various tasks and datasets are impressive, surpassing SOTA metric learning methods by clear margins.

**Strengths:**

+ Interesting idea of constraining the "second-order" feature difference. The intuition of the proposed regularization is natural and the authors provided conceptual experiments to corroborate the point.

+ Impressive umerical results and extensive and informative experiments. I also appreciate the authors efforts for showing the stability of the proposed regularization technique in terms of hyperparameters. This is rare among many deep learning methods, making it much more practical to use in real world.

+ This paper is clearly written.

**Weaknesses:**

- The only concern I have is about the theoretical analysis, especialy thm 3 and 4. The two theorems assumed that $x$, $z$, $\hat{x}$ and $\hat{z}$ are from the same distribution. I wonder if the assumption is so strong that makes the result meaningless in real world. The authors are clearly more interested in regularizing the differences of inter-class feature difference. In that case the samples are clearly from different distributions, as the authors assumed in the conceptual experiments in Fig. 2.

I am concerned that the math analysis is only to make the paper more "mathy" and of less practical meaning. I would love to raise the score if the authors could help to clarify or tone down the theoretical analysis. I think this work is good even as a pure empirical one.

**Questions:**

See the weaknesses part.

---

> ### Author Response · Authors · 2023-11-17
> **Response to Reviewer SsFX**
>
> Thanks for your positive and insightful comments! Our explanation and clarification can be found as follows.
>
> ---
>
> **Comment_1:** The only concern I have is about the theoretical analysis, especially thm 3 and 4. The two theorems assumed that $x$, $z$, $\widehat{x}$, and $\widehat{z}$ are from the same distribution. I wonder if the assumption is so strong that makes the result meaningless in the real world. The authors are clearly more interested in regularizing the differences of inter-class feature differences. In that case, the samples are clearly from different distributions, as the authors assumed in the conceptual experiments in Fig. 2.
>
> **Response_1:** Thank you for the detailed comment! Please allow us to clarify that in each sub-figure of Fig. 2, all three different clusters are considered to follow a single overall distribution with mixed Gaussians (i.e., each sub-figure is a *Gaussian mixture model* (GMM)), and thus both the training and test examples are sampled from such a single distribution (as we had discussed in Appendix A.1). In this case, the data points $x$ and $\widehat{x}$ (as well as $z$ and $\widehat{z}$) may come from the same cluster or from two different clusters, but they are sampled from the same distribution of the corresponding GMM.
>
> We totally agree with the reviewer that the *independent and identically distributed* (i.i.d.) assumption is indeed very important for our theoretical analyses. However, here the i.i.d. assumption on the data points $x$, $z$, $\widehat{x}$, and $\widehat{z}$ is actually a relatively mild condition that has been commonly applied in many classical machine learning tasks. The conventional analyses [R1-R3] on generalization error are also based on the i.i.d. assumption, so we simply follow this common practice to perform our theoretical analyses in Theorem 3 and 4.
>
> In summary, our i.i.d. assumption in Theorem 3 and Theorem 4 is generally mild, and it is also consistent with the practical setting of our experiments. Therefore, we think that the analytical results are still meaningful in real-world applications.
>
> Reference
>
> [R1]. Generalization Bounds for Metric and Similarity Learning,  MLJ’16.
>
> [R2]. A Theoretical Analysis of Contrastive Unsupervised Representation Learning, ICML’19.
>
> [R3]. Curvilinear Distance Metric Learning, NeurIPS’19.

---

### Author Response · Authors · 2023-11-20
**Dear Reviewers**

Dear Reviewers,

Thank you very much for dedicating your time to reviewing our paper.

We are reaching out to express our concerns regarding the rebuttal process.

If you have any additional questions after reviewing our responses, we are glad to provide targeted answers to address your further concerns.

Kind regards,

Authors of Paper ID 2934

---

### Meta-Review · Area_Chair_kFAo · 2023-12-06

**Metareview:**

This paper introduces a novel difference alignment regularization technique (DAR) for similarity learning that encourages consistency between inter-class representation differences. Reviewers recognize the natural intuition and effectiveness of this idea both theoretically and empirically. Results surpass prior metric learning methods by clear margins across multiple domains including image retrieval, face recognition, and document classification.

The consistently strong appraisal across other dimensions supports acceptance. Authors are encouraged to address theoretic inconsistencies, and position better versus prior arts.

**Justification For Why Not Higher Score:**

Primary concerns centered around the theoretical assumptions and analysis. Specifically, relying on first-order feature distance metrics in the bounds seems contradictory to the higher-order motivation. Reviewers also ask for more comparisons to related ideas like contrastive losses and inter-class compactness techniques. Additionally, one reviewer found notation and theorem statements unnecessarily complex.

**Justification For Why Not Lower Score:**

Multiple reviewers praised the extensive experiments validating performance gains and hyperparameter sensitivity. The simplicity and stability of DAR enhance its practical appeal for real-world usage.

---

### Decision · Program_Chairs · 2024-01-16

Accept (poster)